# Measurement report: Long-term variations in surface NO$_X$ and SO$_2$ mixing ratios from 2006 to 2016 at a background site in the Yangtze River Delta region, China

Qingqing Yin [1], Qianli Ma[2], Weili Lin[1], Xiaobin Xu[3], Jie Yao[2]

[1] Key Laboratory of Ecology and Environment in Minority Areas (Minzu University of China), National Ethnic Affairs Commission, Beijing 100081, China
[2] Lin'an Atmosphere Background National Observation and Research Station, Lin'an 311307, Hangzhou, China
[3] Key Laboratory for Atmospheric Chemistry, Chinese Academy of Meteorological Sciences, Beijing 100081, China

*Correspondence to*: Weili Lin (linwl@muc.edu.cn)

**Abstract.** China has been experiencing rapid changes in emissions of air pollutants in recent decades. Increased emissions of primary particulates and reactive gases caused severe haze in several polluted regions including the Yangtze River Delta (YRD). Measures implemented in recent years for improving air quality have reduced the emissions of NO$_X$, SO$_2$, etc. The emission changes of these gases are reflected by tropospheric columns from satellite observations and surface measurements of surface concentrations from urban sites. However, little is known about the long-term variations in regional background

NO$_X$ and SO$_2$. In this study, we present NO$_X$ and SO$_2$ measurements from the Lin'an station (LAN, 119°44′ E,30°18′ N,138.6 m a.s.l.), one of the Global Atmosphere Watch (GAW) stations in China. We characterize the seasonal and diurnal variations and study the long-term trends of NO$_X$ and SO$_2$ mixing ratios observed at LAN from 2006 to 2016. We also interpret the observed variations and trends in term of changes in meteorological conditions as well as emission of these gases. The overall average mixing ratios of NO$_X$ (NO$_2$) and SO$_2$ during 2006–2016 were 13.6 ± 1.2 ppb (12.5 ± 4.6) and 7.0 ± 4.2 ppb, respectively.

The averaged seasonal variations showed maximum values of NOx and SO$_2$ in December (23.5 ± 4.4 ppb) and January (11.9 ± 6.2 ppb), respectively, and minimum values of 7.1 ± 0.8 ppb and 2.8 ± 2.3 ppb (both in July), respectively. The average diurnal variation characteristics of NO$_X$ and SO$_2$ differed considerably from each other though the daily average mixing ratios of both gases were significantly correlated ($R^2 = 0.29$, $P < 0.001$). The annual average mixing ratio of NO$_X$ increased during 2006–2011 and then decreased significantly at 0.78 ppb/yr (–5.16 %/yr, $P < 0.01$). The annual 95 % and 5 % percentiles of

hourly NO$_X$ mixing ratios showed upward trends until 2012 and 2014, respectively, before a clear decline. The annual average mixing ratio of SO$_2$ decreased significantly at 0.99 ppb/yr (–8.27 %/yr, $P < 0.01$) from 2006–2016. The annual 95 % and 5 % percentiles of hourly SO$_2$ mixing ratios all exhibited significant ($P < 0.001$) downward trends at 3.18 ppb/yr and 0.19 ppb/yr, respectively. Changes in the total NO$_X$ and SO$_2$ emissions as well as the industrial emissions in the YRD region were significantly correlated with the changes in annual NO$_X$ and SO$_2$ mixing ratios. The significant decreases in NO$_X$ from 2011

to 2016 and SO$_2$ from 2006 to 2016 highlight the effectiveness of relevant control measures on the reduction in NO$_X$ and SO$_2$ emissions in the YRD region. A decrease of annual SO$_2$/NO$_X$. ratio was found, suggesting a better efficacy in the emission reduction of SO$_2$ than NO$_X$. We found gradual changes in average diurnal patterns of NO$_X$ and SO$_2$, which could be attributed

to increasing contributions of vehicle emissions to $NO_X$ and weakening impacts of large sources on the $SO_2$ concentration. This study reaffirms China's success in controlling both $NO_X$ and $SO_2$ in the YRD but indicate at the same time a necessity to
strengthen the $NO_X$ emission control.

**Keywords**: background $NO_X$ and $SO_2$; long-term trend; emission reduction.

## 1 Introduction

China's economy has experienced decades of rapid development, resulting in considerable pollutant emissions from coal combustion and motor vehicles, which affect ambient air quality and human health (Kan et al., 2009, 2012; Liang et al., 2019).
$NO_X$ and $SO_2$ are two major gaseous pollutants that are essential precursors to secondary aerosol formation and acidification (Li et al., 2020). Therefore, the changes in $NO_X$ and $SO_2$ emissions have been receiving increasing attention in China (Zhao et al., 2013; Zhao et al., 2018). To improve air quality, the Chinese government has promulgated a series of policies and regulations on $SO_2$ and $NO_X$ control, especially since 2006 and 2011, respectively (Zheng et al., 2015).

Long-term observations of $NO_X$ and $SO_2$ are not only critical for the integrated assessment of air quality and atmosphere–
biosphere interactions (Swartz et al., 2020a), but also for the analysis of their reduction effects on $PM_{2.5}$, nitrate, sulphate, and near-surface $O_3$, providing a basis for further improvement of atmospheric protection policies (Yu et al., 2019). At a regional scale, long-term, reliable $NO_X$ and $SO_2$ observations can also provide data to enable the scientific community to predict the future state of the atmosphere and assess environmental policies, serving to reduce environmental risks and enhance climate, weather, and air quality prediction capabilities (GAW, 2017). Numerous studies have evaluated the effectiveness of $NO_X$ and
$SO_2$ control in China from a long-term perspective by using emission inventories, satellite retrieval data, and ground monitoring data. For example, Sun et al. (2018) used a unified source emission inventory approach to quantify the historical emission trends of $SO_2$ and $NO_X$ in China from 1949 to 2015; the results indicated that these pollutants reached an inflection point in 2006 and 2011, respectively. Source emission inventories by Kurokawa and Ohara (2020) revealed similar patterns. During the period from January 2005 to December 2015, the column concentration of $NO_2$ from ozone monitoring instrument (OMI)
satellite retrieval indicated an increasing trend in most of China until a gradual or slight decrease in 2011 or 2012 (Cui et al., 2016). Zhao et al. (2019) used ground-based $NO_2$ observations to assess the effectiveness of pollution control policy in a southwestern city cluster and revealed fluctuations in $NO_2$ mixing ratios from 2008 to 2013, followed by an irregular declining trend after 2013. All these studies reported that $NO_X$ and $SO_2$ mixing ratios have been effectively controlled in China despite the increasing economic development over the past decades.
The Yangtze River Delta (YRD) region is located in the central-eastern region of China, which has the largest economic output in China and has the sixth largest urban agglomeration in the world. The region covers an area of 359,100 $km^2$ and has a population of 224 million, accounting for 16.08 % of the country's population (Fang and Tian, 2020). Because of increases in population, urbanization, and industrialization in recent decades, the air pollution in the YRD has exhibited complex and regional characteristics (Li et al., 2019; Wang et al., 2019), and the YRD has become one of the most polluted regions in the

world (Xie, 2017b), with $NO_X$ and $SO_2$ being the main factors that influence air quality in the region (Yang and Luo, 2019). Xu et al. (2008) compared observational data in 2005–2006 with those 10 years earlier and concluded that as early as the mid-1990s, $SO_2$ and $NO_X$ mixing ratios had already become considerably high at the background station in the YRD; since then, anthropogenic emissions have caused a substantial increase in the $NO_X$ concentration, making $NO_X$ another major pollutant in addition to $SO_2$. The implementation of pollution control policies and continual innovation in $SO_2$ pollution control technology

have mitigated $SO_2$ pollution in the YRD, resulting in a consistent decrease in $SO_2$ mixing ratios (Qi et al., 2012); however, $NO_X$ mixing ratios remain high (Shi et al., 2018).

In this paper, we present 11-year (2006–2016) surface $NO_X$ and $SO_2$ observation data from Lin'an regional atmospheric background station. We analysed the long-term variations of $NO_X$ and $SO_2$ and their influencing factors in the YRD background area to (1) assess the effectiveness of pollution control in the area and (2) provide a scientific basis and reference for future

pollution control strategies.

## 2   Information and methods

### 2.1 Site information

The Lin'an regional atmospheric background monitoring station (119°44′ E, 30°18′ N, 138.6 m a.s.l.; referred to LAN) is located in Lin'an District, Hangzhou City, Zhejiang Province (Fig. 1) and is one of the regional atmospheric background

stations operated by China Meteorological Administration; it is also a World Meteorological Organization (WMO) Global Atmospheric Watch (GAW) member station. LAN is located on an isolated hilltop, surrounded by hilly and mountainous terrain, with no large villages within a 3 km radius. It is within the region of subtropical monsoon climate, with the most dominant wind direction from the northeast and the secondary from the southwest. The seasonal variations in meteorological elements, namely atmospheric pressure (P), temperature (T), wind speed (WS), relative humidity (RH), and rose maps of wind

speed (WS) and wind direction frequency (WF), are presented in Fig. 2.

### 2.2 Observations and quality control methods

At the LAN station, observations of $O_3$, $NO_X$, $SO_2$, and CO are performed by an integrated observation and quality control system combining $O_3$, $NO_X$, $SO_2$, and CO analysers, calibration equipment, and ancillary materials, such as standard gases and zero air supply (Lin et al., 2009). $NO_X$ and $SO_2$ were measured using a Model 42C-TL trace-level chemiluminescent analyser

and a Model 43C-TL trace-level pulsed fluorescence analyser (Thermo Fishier Scientific, MA, USA), respectively. In Model 42C-TL trace-level chemiluminescent analyser, $NO_2$ is converted to NO by a molybdenum $NO_2$-to-NO converter heated to about 325°C. The converter efficiency was checked annually using gas phase titration (GPT). If the converter efficiency is less than 96%, replace the converter. Data are recorded as 5 min averages. The meteorological parameters (WS, wind direction, T, and RH) for a given period were obtained from the routine meteorological observations at the station. The main objective of

operational observations of reactive gases at regional background stations is to obtain accurate trends in the measured reactive gases, for which reliable and comparable data are essential. Therefore, strict quality control measures were implemented during the observation process (Lin et al., 2019). The quality control measures mainly included the following: (1) daily zero and span checks (automatic); (2) monthly multi-point calibrations (≥5 points, including zero); (3) comparisons of reference $SO_2/N_2$ and $NO/N_2$ gas mixtures to the standards of the National Institute of Standards and Technology before and after their usage (periodically) to ensure data traceability; (4) instrument self-diagnosis, manual testing, checking, and maintenance (US EPA, 2017); and (5) data correction according to the quality control results, especially the results of zero/span checks and multipoint calibrations.

From 1 January 2006 to 31 December 2016, a total of 93,759 and 90,453 valid hourly average data points were obtained for $NO_X$ and $SO_2$, respectively. Missing data totalled 2673 h and 5979 h for $NO_X$ and $SO_2$, respectively. The missing $NO_X$ data were mainly for the period from 2 to 13 February 2007 and from 24 July to 8 October 2012. The missing $SO_2$ data were mainly for the period from 23 September to 21 December 2013, from 8 to 26 May 2014, and from 17 October 2014 to 24 January 2015.

**2.3 Data processing methods**

(1) Data statistics. The daily means of $NO_X$ and $SO_2$ were calculated using the hourly average data, and only daily mean data calculated from at least 18 hourly data were used as valid daily means. The monthly means of $NO_X$ and $SO_2$ were calculated from the valid daily average data and considered valid if they were based on at least 21 valid daily averages (or at least 17 valid daily averages in February). Annual means were calculated on the basis of the complete monthly mean data each year. If a month's mean data were unavailable, we used an interpolating value from the corresponding monthly means in different years during the observation. In China, spring is from March to May, summer is from June to August, autumn is from September to November, and winter is from December to February.

(2) Monthly satellite based $NO_2$ OMI data were provided by Lin's research group at Peking University; the data were retrieved using an optimized inversion algorithm (Lin et al., 2014; Lin et al., 2015; Boersma et al., 2019). A grid range of 115.125° E–122.875° E and 27.125° N–35.875° N was selected to cover the entire YRD region.

**2.4 Concentration weighted trajectory method**

We used the concentration weighted trajectory (CWT) method to identify potential source areas (PSAs) of $NO_X$ and $SO_2$ because this method can effectively distinguish the relative strength of potential sources (Xin et al., 2016). In the CWT method, the study area is divided into $i \times j$ small grids with equal size, and each grid (i, j) is assigned a weighted concentration according to the following equation:

$$C_{ij} = \frac{1}{\sum_{k=1}^{m} \tau_{ijk}} \sum_{k=1}^{m} C_k \, \tau_{ijk} \qquad (1)$$

Where $k$ denotes the indicator of a trajectory, $m$ denotes the total number of trajectories, $C_k$ denotes the concentration observed when trajectory $k$ arrives, and $\tau_{ijk}$ is the residence time of trajectory $k$ in the $ij_{th}$ grid cell. To reduce errors in the more distant grids, an empirical weighting factor $W_{ij}$ is introduced (Wang et al., 2006; Deng et al., 2020), with the following equation:

$$\text{CWT}(i,j) = W_{ij} \times C_{ij} \qquad (2)$$


$$W_{ij} = \begin{cases} 1 & (\, n_{i,j} > 3n_{ave} \,) \\ 0.7 & (\, 3n_{ave} < n_{i,j} < 1.5n_{ave} \,) \\ 0.42 & (1.5n_{ave} < n_{i,j} < n_{ave}) \\ 0.05 & (\, n_{i,j} < n_{ave} \,) \end{cases} \qquad (3)$$

Here,

$$n_{ave} = \frac{D \times t \times n}{i \times j} \qquad (4)$$

Where $D$ denotes the number of days included, $t$ denotes the number of trajectories per day, $n$ denotes the trajectory endpoints

of each trajectory, and $i \times j$ denotes the total number of grids.

We used a hybrid single-particle Lagrangian integrated trajectory model (Hysplit4.9) from National Oceanic and Atmospheric Administration, USA, to calculate the 24-h backward trajectories at 10 m above ground level over LAN during 2006–2016; the NCEP–NCAR reanalysis meteorological data set (https://ready.arl.noaa.gov/archives.php) and was used to calculate the trajectories and atmospheric mixed layer heights. The computed backward trajectories were subsequently processed using the

TrajSat plug-in for CWT in Meteoinfo software (Wang, 2014), covering the region located within 20–40° N and 110–130° E and with a grid size resolution of 0.5 ° × 0.5 °.

## 3 Results and discussion

### 3.1 Observational levels and comparison with other sites

The hourly average $SO_2$ mixing ratios ranged from 0.1 ppb to 128.6 ppb, which were all below the GB3095–2012 secondary

standard limit for $SO_2$ (190 ppb). The hourly average $NO_X$ mixing ratios at LAN ranged from 0.4 ppb to 165.6 ppb, with $NO_2$ mixing ratios ranging from 0.2 ppb to 106.8 ppb. Only 3 hours' data exceeded the secondary standard limit value for $NO_2$ (106 ppb) as stated in the national ambient air quality standard (GB3095–2012). It should be mentioned that the measurement of $NO_2$ was via conversion to NO by a molybdenum $NO_2$-to-NO converter heated to about 325 °C, which was known to suffer from the interference of other NOy compounds such as PAN and $HNO_3$ (Steinbacher et al., 2007; Jung et al., 2017). This

implies that the measured $NO_2$ mixing ratios were higher than actual values. However, it is impossible to quantify the overestimated parts due to the lack of other information. The interference might be enhanced with the increasing ratios of PAN

to NOx (PAN/NO$_X$). Qiu et al. (2020) reported an increasing PAN/NO$_X$ from 2011 to 2018 at a background site in the North China Plain, but it is not clear if there was a similar increase in PAN/NO$_X$ in the YRD. During the transport of air masses to the background site, HNO$_3$ should be reduced by deposition or partitioning in the particulate phase and intercepted by filters before NOx was measured. Since NOz (NOy-NOx) was produced by NOx oxidation, the overestimation of NOx by partial conversion of NOz, in turn, might be a positive offset in the difference between the measured mixing ratios and the emission of NOx when discussing their long-term trends.

Table 1 presents annual statistics of the NO$_2$, NO$_X$ and SO$_2$ mixing ratios observed at LAN between 2006 and 2016. The overall average mixing ratios with ± 1 standard deviation of for NO$_X$ (NO$_2$) and SO$_2$ from 2006 to 2016 were 13.6 ± 1.2 ppb (12.5±4.6 ppb) and 7.0 ± 4.2 ppb, respectively, with the highest NO$_x$ (NO$_2$) value being observed in 2012 and the highest SO$_2$ in 2006. NO$_2$ was the dominant form of NO$_X$, accounting for 82.2 % of NO$_X$ (according to the slope value from the reduced major axis regression on hourly average NO$_2$ and NO$_X$ data). The average NO$_2$ mixing ratio was 12.5 ± 4.6 ppb, which was below the primary annual limit of 21.2 ppb in GB 3095–2012. Some information on NO$_2$(NO) can be seen the supplementary material (Tab.S1). The average SO$_2$ mixing ratio from 2006 to 2016 is close to the primary annual limit of 7.6 ppb in GB3095–2012. However, the annual average SO$_2$ mixing ratios (10.6–14.6 ppb) from 2006 to 2008 was much higher than the limit of the primary standard though lower than the limit of the secondary standard (22.8 ppb).

Table 2 compares the levels of NO$_X$ and SO$_2$ mixing ratios at LAN with those corresponding SO$_2$/NO$_X$ ratios at other background stations in seven geographic regions of China: north, east, south, northeast, northwest, southwest, and central China. The NO$_X$ mixing ratio at LAN was slightly higher than that at Shangdianzi (12.7 ± 11.8 ppb) in northern China, equal to that at Dinghushan (13.6 ppb) in southern China, and much higher than those at Wuyishan (2.70 ppb) in eastern China, Fukang (8.3 ppb) in northwest China, Changbai Mountain (4.7 ppb) in northeast China, Jinsha (5.6 ± 5.5 ppb) in central China, and Southwest Gongga Mountain (0.90 ppb). These results indicate that LAN recorded the highest level of NO$_X$ among the regional atmospheric background stations in China, which could be attributed to the developed economy of the YRD region. The SO$_2$ mixing ratio at LAN was close to that at Shangdianzi (7.6 ± 10.2 ppb) in northern China, higher than that at Dinghu Mountain (6.5 ppb) in southern China, and much higher than those at Wuyishan (1.48 ppb) in eastern China, Changbai Mountain (2.1 ppb) in northeast China, Fukang (2.2 ppb) in northwest China, Gongga Mountain (0.19 ppb), and Jinsha (2.8 ± 5.5 ppb) in central China. The regional difference in NO$_X$ and SO$_2$ was closely related to the diverse levels of economic development in China's regions because it was broadly characterised by a higher level in the eastern than in central and western regions. The SO$_2$/NO$_X$ ratio at LAN was at a high level in China, which reflects the different energy structures to some extent.

**3.2 Seasonal variations**

Figure 3 illustrates the average seasonal variations in NO$_X$ and SO$_2$ mixing ratios at LAN. The maximum monthly average mixing ratios of NO$_X$ and SO$_2$ were observed in December and January, at 23.5 ± 4.4 ppb and 11.9 ± 6.2 ppb, respectively. The minimum values both occurred in July, at 7.1 ± 0.8 ppb and 2.8 ± 2.3 ppb, respectively. The average monthly variations in NO$_X$ exhibited significant correlations with the monthly NO$_2$ satellite data ($R^2 = 0.82$, $P < 0.001$). Seasonal variation patterns

of NO$_X$ and SO$_2$ look alike, showing a concave shape with its minimum in summer. The highest mixing ratios occurred in winter (NO$_X$: 19.5 ppb; SO$_2$: 10.1 ppb), followed by spring (NO$_X$: 13.4 ppb; SO$_2$: 7.8 ppb), autumn (NO$_X$: 13.6 ppb; SO$_2$: 6.7 ppb), and summer (NO$_X$: 8.1 ppb; SO$_2$: 3.3 ppb). The monthly average mixing ratios of both NO$_X$ and SO$_2$ showed a dip in February—a phenomenon also observed in NO$_X$ and SO$_2$ (Wang et al., 2016; Xue et al., 2020) and NO$_3^-$ and SO$_4^{2-}$ in PM$_{2.5}$ in Shanghai (Duan et al., 2020). The source emission inventory data indicated that NO$_X$ and SO$_2$ emissions from industry,

transportation, and coal-fired power plants were all lower in February than in January and March throughout China (Li et al., 2017), which may be related to decreased emissions due to lower economic activity during Chinese Spring Festival. In addition, the higher RH in February (Fig. 2) might have led to higher NO$_X$ and SO$_2$ removal rates.

**3.3 Diurnal variations**

Figure 4 shows the annual and seasonal average diurnal variations in NO$_X$ and SO$_2$ at LAN from 2006 to 2016, along with the

annual average diurnal variations in NO$_X$ and SO$_2$ at some other sites in the YRD. The overall diurnal profile of NO$_X$ displayed a double-peak and double-valley pattern (Fig. 4a). The valley values occurred at 05:00–06:00 and 13:00, with mixing ratios of 12.3 ppb and 10.0 ppb, respectively, and the peak values occurred at 09:00 and 19:00, with mixing ratios of 13.1 ppb and 14.4 ppb, respectively. Surrounding areas, such as Chongming, Pudong (Xue et al., 2020), and Xujiahui (Gao et al., 2017) in Shanghai City, Hangzhou (Zhou et al., 2020) in Zhejiang Province, and Nanjing (Wang et al., 2017) in Jiangsu Province also

exhibited a double-peak and double-valley type of average diurnal variation in NOx (Fig. 4a), indicating a regional NO$_X$ pollution characteristic. However, at most atmospheric background stations, the average diurnal variations in NO$_X$ exhibited a single-peaked and single-valley pattern, such as those at Xinglong in north China (Yang et al., 2012), Tianhu in the Pearl River Delta (Shen et al., 2019), Dae Hung in South Korea (Pandey et al., 2008), and Mount Cimone in Italy (Cristofanelli et al., 2016), suggesting a more complex anthropogenic influence in the YRD region. In summer, the seasonal average diurnal

variation in NO$_X$ showed a morning peak at 08:00, which time is 1 to 2 h earlier than that occurred in other seasons (Fig. 4c). SO$_2$ at LAN showed relatively small average diurnal variation (Fig. 4b), with higher mixing ratios from midnight to noontime and lower ones during later afternoon and evening. The average diurnal amplitude of SO$_2$ at LAN was much smaller than those found in Nanjing and Jiaxing. The seasonal average diurnal profiles of SO$_2$ at LAN were similar to the annual average one except for that in winter, which had a peak around noon (Fig. 4d).

The diurnal variation of pollutants emitted at ground level are closely related to the intensity of emissions, atmospheric transport, diurnal development in boundary layer height, and atmospheric photochemical reactions (Resmi et al., 2020). The mixing layer depth (MLD) was much lower at night than during the daytime, as shown in Fig. 4b. Low MLDs at night are not conducive to pollutant dispersion, whereas high MLDs during the daytime are conducive to pollutant dispersion. This day-night difference in the MLD is one of the factors causing lower levels of SO$_2$ and NO$_X$ during afternoon hours. Photochemistry

during the daytime also contributes to rapid chemical transformation of SO$_2$ and NO$_X$, which results in low NO$_X$ and SO$_2$ mixing ratios in the afternoon. Overall, the morning peak of NO$_X$ was lower than the evening peak, the morning peak of SO$_2$ was higher than the evening subpeak, and the morning peak of SO$_2$ was not as protruding as and occurred slightly later than

that of $NO_X$, reflecting the differences in their sources. The morning peak of $NO_X$ may be influenced by vehicle emissions during the morning rush hour, and the early peak of $SO_2$ may be more influenced by vertical changes during the developing mixed layer depth height (Qi et al., 2012). The evening peaks of $NO_X$ and $SO_2$ were relatively similar because both were closely related to the MLD decrease and for $NO_X$ likely also vehicle emissions during the evening rush hour.

## 3.4 Influence of meteorological factors

Changes in meteorological factors have considerable effects on the levels of air pollutants. In this section, we investigate the influences of meteorological factors on the variations in $NO_X$ and $SO_2$ mixing ratios through statistical plots showing relationships between pollutant concentrations and meteorological factors as well as correlation analysis. The variation characteristics of hourly average mixing ratios of $NO_X$ and $SO_2$ along with meteorological parameters are presented in Fig. 5. The data are grouped into three subsets corresponding to time periods: I (2006–2009), II (2010–2013) and III (2014–2016). The variation characteristics of $NO_X$ and $SO_2$ with WS (Fig. 5a,b) were consistent during period I, showing decreases of $NO_X$ and $SO_2$ with increasing WS. Higher WS facilitated the dilution of $NO_X$ and $SO_2$ and vice versa. However, the situation for $SO_2$ was different during period II and III, when the $SO_2$ level was stable with the change of WS. The correlation of T between the two pollutants varied considerably, with the $SO_2$ mixing ratios decreasing nearly monotonically with increasing T (Fig. 5d), whereas $NO_X$ increased with increasing T in the low temperature range and decreased with increasing T in the high temperature range (Fig. 5c). Fig. 5c indicates a positive correlation between $NO_X$ and T in winter and negative correlations in other seasons, but the positive correlation in winter is weak and insignificant (Table 3). Pandey et al. ( 2008) reported that low T might facilitate the increase of $NO_X$ emissions from motor vehicle exhaust. The variations in $NO_X$ and $SO_2$ with RH (Fig. 5e,f) exhibit a convex pattern and the former patterns in 3 different periods are well consistent but the latter ones are not at low RH. The correlation between $SO_2$ and RH was stronger than that of $NO_X$ and RH (Table 3). The variation characteristics of $NO_X$ and $SO_2$ mixing ratios with the MLD exhibited diverse patterns (Fig. 5g,h). The mixing ratio of $NO_X$ decreased with increasing MLD. However, the $SO_2$ levels during period II and III remained nearly stable in the whole MLD range and a slight decline of $SO_2$ with increasing MLD was only observed during period I. The difference in $NO_X$ and $SO_2$ mixing ratios with the MLD implies that the $NO_X$ sources mostly impacting the LAN site should be mainly in the near-surface layer, such as emissions from motor vehicles and small burners, whereas $SO_2$ may originate from the vertical exchange of elevated sources transported in the higher altitude layer (200–1300 m).

Figure 6 displays the rose diagrams of $NO_X$ and $SO_2$ mixing ratios in different seasons. There are some seasonal differences in the dependence of $NO_X$ and $SO_2$ on wind direction. In summer, the high mixing ratios of $NO_X$ and $SO_2$ were mainly from the NW–NNE and SSW–NW sectors, respectively (Fig. 6b). In other seasons, relatively high $NO_X$ and $SO_2$ values were mainly from the N–ENE and S–WSW directions, respectively, under the influences of the dominant and subdominant WDs (Fig. 2b, d). Overall, $NO_X$ and $SO_2$ observed at LAN originated mainly from the NW-ENE and SSW-NW sectors, respectively. However, this result provides only little information about the actual geographic distributions of major $NO_X$ and $SO_2$ sources influencing LAN. Therefore, we used the CWT method to identify the PSAs for $NO_X$ and $SO_2$. Fig. 7 presents the areas, from which $NO_X$

and $SO_2$ observed at LAN originated. Although the PSAs covered the entire YRD, the PSAs for the highest $NO_X$ and $SO_2$ levels appeared mainly in the eastern coastal region, which is closely related to the booming local economy. More obvious provincial differences were observed in a higher PSA for $NO_X$ than that for $SO_2$. Temporally, the high PSA (>10 ppb) of $NO_X$ and $SO_2$ was most extensive in winter, followed by spring and autumn, with the least extensive PSA in summer. The $NO_X$ PSAs over coastal areas were more extensive than those for $SO_2$ in each season. The YRD is one of the five major port clusters in China; thus, this region's ship emissions might be a major cause of this difference (Fan et al., 2016; Wan et al., 2020). The CWT analysis indicated that $SO_2$ was mainly influenced by industrial emissions from inland areas, whereas $NO_X$ was mainly influenced by both inland and marine traffic.

## 3.5 Long-term variations in NOx and SO₂ mixing ratios

Fig. 8 displays the variations in the annual and seasonal average $NO_X$ and $SO_2$ mixing ratios observed at LAN during 2006-2016, together with estimated annual emissions in the YRD. The annual average of $NO_X$ showed an increase followed by a decrease, while that of $SO_2$ experienced a nearly monotonic decrease. The annual $NO_X$ mixing ratio revealed an increase, with a rate of +0.31 ppb/yr ($R^2 = 0.28$, $P = 0.16$) during 2006–2011 and a significant decreasing trend, with a rate of −0.78 ppb/yr or −5.16 %/yr ($R^2 = 0.85$, $P < 0.01$) during 2011–2016 (Fig. 8a). The decreasing rate was less than that found in urban Shanghai (-2.1 ppb/yr; Gao et al., 2017). Selecting 2006 as the base year, we compared the annual percentage change in $NO_X$ at LAN (-0.49 %/yr) during 2006–2016 with those of other regions over the same period. The *Ecological and Environmental Status Bulletin* (Shanghai Municipal Bureau of Ecology and Environment, 2007–2017; Department of Ecology and Environment of Zhejiang Province, 2007–2017; Department of Ecology and Environment of Jiangsu Province, 2007–2017) reported a similar change of −0.45 %/yr in the YRD region (without data for Anhui Province), reflecting the suitable regional representativeness of LAN. The annual percentage decrease of $NO_X$ at LAN and in the YRD was much smaller than those in many regions—for example, the Pearl River Delta in China (−2.84 %/yr; Yan et al., 2020), Kraków City in Poland (−2.21%/yr; Agnieszka and Gruszecka-Kosowska, 2020), at Preila station in Lithuania (−1.60 %/yr; Davuliene et al., 2021), and in New York City in the United States (−3.46 %/yr; Squizzato et al., 2018)—but more favourable than those in some other regions, such as Wuhan City in China (+2.08 %/yr; Li et al., 2020) and Amersfoort City (+6.50 %/yr) and Louis Trichardt City in South Africa (+1.85 %/yr; Swartz et al., 2020b). Compared with other background regions in China, the annual change of $NO_X$ at LAN was less favourable than that in north China (−3.34 %/yr) with a base year of 2005 (Bai et al., 2015) and more favourable than that in northwest China (+12.98 %/yr) with a base year of 2010 (Li et al., 2019).

Figure 8 also presents the NOx emission data from the *China Ecological Environment Bulletin* in different years. The change of the annual average NOx  mixing ratio was highly correlated with the total $NO_X$ emissions ($R^2 = 0.92$, $P < 0.001$) and total industrial emissions ($R^2 = 0.94$, $P < 0.001$) in the YRD region. The peak surface $NO_X$ mixing ratio was observed in 2011. Since China began to control and reduce $NO_X$ emissions as part of the *12th Five-Year Plan* (2011–2015) and promulgated the strict *Air Pollution Prevention and Control Action Plan* in 2013, many flue gas denitrification systems have been installed in coal-fired power plants and heavy industry operations (Zhao et al., 2019), resulting in a decrease in annual $NO_X$ emission since

2011. As seen in Figure 8a, the total and the industrial NOx emission showed increasing trends with 5.84 %/yr ($R^2 = 0.91$, $P =$ 0.011) and 6.3%/yr ($R^2 = 0.91$, $P = 0.006$), respectively, from 2007-2011, with $-7.63$%/yr ($R^2 = 0.91$, $P = 0.003$) and $-8.30$%/yr ($R^2 = 0.84$, $P = 0.011$), respectively, from 2011-2016. The seasonal long-term trends of $NO_X$ did always resemble the annual trend. While seasonal $NO_X$ mixing ratios in winter, autumn, and spring increased before 2011 and then decreased, just like the annual $NO_X$ mixing ratio did, the seasonal $NO_X$ mixing ratio in summer exhibited a nearly monotonic decreases from 2006 to 2016 at 0.11 ppb/yr ($R^2 = 0.20$, $P = 0.09$) (Fig. 8c). Regarding the seasonal linear fitting trends, the highest increasing and declining trends were observed in winter ($+1.29$ ppb/ yr, $R^2 = 0.52$, $P = 0.06$; $-2.33$ ppb/yr, $R^2 = 0.94$, $P < 0.01$), followed by autumn ($+1.24$ ppb/yr, $R^2 = 0.65$, $P = 0.02$; $-0.41$ ppb/yr, $R^2 = 0.12$, $P = 0.30$) and spring ($+0.31$ ppb/yr, $R^2 = 0.93$, $P < 0.001$; $-1.16$ ppb/yr, $R^2 = 0.76$, $P = 0.09$). We found a significant correlation ($P < 0.05$) between surface $NO_2$ mixing ratio and OMI $NO_2$ vertical column density over YRD (Fig. S3b). To better compare the changes in the two over the same period, we have fitted a linear fit to the data from 2006 to 2011 and from 2011 to 2016 respectively (Fig.S3a). The surface and the OMI $NO_2$ increased at 2.23%/yr ($R^2 = 0.264$, $P = 0.17$) and 5.87%/yr ($R^2 = 0.855$, $P < 0.01$) (based on 2006), respectively, during the up period and decreased at $-4.98$%/yr ($R^2 = 0.823$, $P < 0.01$) and $-4.22$%/yr ($R^2 = 0.897$, $P < 0.01$), respectively, during the declining period.

Annual mean $SO_2$ mixing ratios revealed a significant decreasing trend ($-0.99$ ppb/yr, $R^2 = 0.92$, $P < 0.001$) during 2006-2016 (Fig. 8b). The annual decreasing rate of $SO_2$ at LAN ($-8.27$ %/yr) was more rapid than those in the whole YRD ($-6.65$ %/yr), in the background area in north China ($-0.78$ %/yr; Bai et al., 2015), and in northwest China ($-5.4$ %/yr; Li et al., 2019). Different from $NO_X$, the annual average of $SO_2$ at LAN decreased more rapidly than in most of the aforementioned regions (Table 4), which demonstrates the effectiveness of the policies in controlling $SO_2$ emission during the observation period in the YRD.

The change in the annual $SO_2$ mixing ratio was closely correlated with changes in thermal power plants $SO_2$ industrial emission ($R^2 = 0.89$, $P < 0.001$), industrial $SO_2$ emission ($R^2 = 0.76$, $P < 0.001$) and total $SO_2$ emission ($R^2 = 0.78$, $P < 0.001$) in the YRD (Fig. 8b). In 2011, the $SO_2$ mixing ratio rebounded slightly, with an increase of 9 % compared with the value in 2010. This seemed to be consistent with the variation of industrial $SO_2$ emission. The weakening impact of the global financial crisis and the recovery of industry in the YRD region may explain this slight rebound in $SO_2$ emissions (Xie, 2017b). Seasonally, the $SO_2$ mixing ratio exhibited the strongest decreasing trend ($-1.69$ ppb/yr, $R^2 = 0.90$, $P < 0.001$) in winter, followed by spring ($-1.05$ ppb/yr, $R^2 = 0.97$, $P < 0.001$) and autumn ($-0.99$ ppb/yr, $R^2 = 0.93$, $P < 0.001$), with the smallest decreasing trend observed in summer ($-0.35$ ppb/yr, $R^2 = 0.61$, $P < 0.001$).

In the annual statistics, the 95th and 5th percentile of the pollutants' concentrations can be regarded as influenced by polluted and clean air masses, respectively. The annual trends of the 95th percentile of $NO_X$ and $SO_2$ (Fig. 9a) exhibited similar patterns to the corresponding trends in annual average mixing ratios (Fig. 8a, b), but the peak of the 95th percentile of $NO_X$ occurred in 2012, instead of in 2011. Hao and Song (2018) noted that the $NO_X$ emissions from vehicles peaked in Hangzhou and Ningbo in 2012, which may explain the peak of the 95th percentile occurring later than that in the annual data. Moreover, the 95th percentile of the $SO_2$ mixing ratio decreased at a remarkable rate ($-8.9$ ppb/yr) from 2007 to 2009, which is approximately 2.8

times as strong as the overall rate of decrease during the 11-year period ($-3.2$ ppb/yr). Substantial decreases were also found in the 95th percentiles of the CO mixing ratio (Chen et al., 2020) and the $NO_X$ mixing ratio from 2007 to 2009 at LAN. It is

highly possible that this phenomenon was caused by reduced industrial productions and related emissions following the 2008 global financial crisis. As displayed in Fig. 9b, the level of $NO_X$ in cleaner air mass arriving at LAN exhibited an increasing trend, with a rate of $+0.17$ ppb/yr, from 2006 to 2014 ($R^2 = 0.86$, $P < 0.001$) and then declined after 2014. This is inconsistent with the trend of the 95th percentile of the $NO_x$ mixing ratio, suggesting the polluted and relative clean air masses arriving at LAN were impacted by different emission sources of $NO_x$. Interestingly, the 5th percentile of the $NO_x$ level was significantly

correlated ($R^2 = 0.74$, $P < 0.001$) with the road emissions of $NO_2$ in the YRD (Kurokawa and Ohara, 2020), suggesting that the lower end of $NO_x$ mixing ratios was mainly determined by long-range transported background air containing $NO_x$ from road emissions, while the high end was mainly associated with emissions from industrial production as well as power generation. The level of $SO_2$ in cleaner air mass exhibited a decreasing trend at a rate of $-0.2$ ppb/yr ($R^2 = 0.61$, $P < 0.01$).

Figure 10 displays the scatter plot of the daily average $SO_2$ and $NO_X$ mixing ratio during period I, II and III at LAN. Reduced

major axis regressions were performed on three data subsets. The daily mean mixing ratios of $NO_X$ were significantly ($R^2 = 0.29$, $P < 0.001$) and positively correlated with those of $SO_2$. The ratios of $SO_2$ to $NO_X$ ($SO_2/NO_X$) were 0.96, 0.53, and 0.33 (slopes in the regression lines) during period I, II and III, respectively. The decreasing $SO_2/NO_X$ suggests that $SO_2$ emissions were more efficiently reduced than $NO_X$ emissions. Such a change in emission ratio not only affected ambient $SO_2/NO_X$ but also the ratios of sulphate/nitrate in $PM_{2.5}$ in Shanghai from 2009 to 2012 (Zhao et al., 2015) $SO_4^{2-}/NO_3^-$ in rainwater in

Hangzhou (Yang, 2018; Xu et al., 2019). These results indicate that $NO_X$ has been gaining a more important role in the processes of precipitation acidification and secondary inorganic aerosol formation in the YRD region. Therefore, $NO_X$ emission reduction should be further strengthened in subsequent air pollution control measures and legislation in the YRD region.

Figure 11 reveals the average diurnal variations in $NO_X$ and $SO_2$ during the period I, II and III. During these three periods, the

average diurnal curves in $NO_X$ exhibited a valley around 13:00, with minimum values of 7.5 ppb, 11.2ppb, and 9.2 ppb, respectively. The morning and evening $NO_X$ peaks, which occurred respectively at 09:00 and 19:00, became increasingly distinct over time (Fig. 11a, c, e). The morning and evening peak $NO_X$ values were 9.8 ppb and 10.9 ppb during period I, 14.6 ppb and 15.8 ppb during period II, and 12.3 ppb and 13.6 ppb during period III. The gradual protruding of the morning and evening peaks should be mainly caused by increasing vehicle emissions during the morning and evening rush hours. According

to the 2010 Annual Report on China's Motor Vehicle Pollution Prevention and Control, the state introduced a series of policies to promote automobile and motorbike ownership in response to the international financial crisis and to ensure economic growth; these policies effectively stimulated the automobile market (Mi and Qin, 2011; Hao and Song, 2018) and led to an increase in vehicle emissions and atmospheric oxidation in the YRD region (Yu et al., 2019). Thus, the $NO_X$ mixing ratios around the morning and evening peaks were much higher than those at night during period II (Fig. 11e), which differs much from the

pattern during period I (Fig. 11a). The disappearance of the small peak around 01:00 at night during 2012–2016 may be related to the introduction of stricter air pollution control policies for factories that emit at night. Small peaks in $NO_X$ and $SO_2$ occurred

between 01:00 and 02:00, which might be related to nighttime emissions from unscrupulous enterprises (Fan et al., 2013) or more production activities with lower electricity prices after midnight in response to the financial pressure of the 2008 economic crisis and the corresponding increase in electricity prices for industrial users (Sun, 2008). In spite of these two reasons, however, it's really hard to tell exactly why these small peaks dominate after midnight.

The average diurnal variation curve of $SO_2$ at LAN period I (Fig. 11b) is of the single-valley type, with an average valley mixing ratio of 6.5 ppb. After 2010, the peak shape has changed from single-valley type to the double-peak and double-valley type (Fig. 11d, f). The valleys of $SO_2$ during period II occurred at 06:00 and 15:00, with average mixing ratios of 5.2 ppb and 4.7 ppb, and the peaks occurred at 10:00 and 19:00, with average mixing ratios of 5.9 ppb and 5.3 ppb, respectively. The $NO_X$ and $SO_2$ evening peaks occurred at the same time (19:00), but the $SO_2$ morning peak time was 1 hour later than the $NO_X$ morning peak (09:00), indicating that the $NO_X$ and $SO_2$ morning peaks were influenced by different sources, whereas the evening peaks were from similar sources. The formation of the $SO_2$ morning peak may be mainly related to the vertical exchange during the development of the atmospheric boundary layer and the air in the upper layer with a higher $SO_2$ mixing ratio than that at the surface draining down. The formation of the evening peaks of $NO_X$ and $SO_2$ may be mainly related to the increase in motor vehicle and residential sources emissions, which are stronger in the rush and cooking hours and that of $SO_2$ may be probably more due to the reduction of power plants emissions. Compared with that during period II, the $SO_2$ mixing ratios at the morning and evening peaks in period III were approximately 3 ppb lower, suggesting that the large emitters that release $SO_2$ all the time were emitting less and less.

**Conclusions**

In this study, we characterized the seasonal and diurnal variations and analysed the long-term trends in $NO_X$ and $SO_2$ mixing ratios in the YRD background area during the period of 2006–2016. We also tried to understand the variations and trends in terms of the changes in emissions and meteorological conditions. The hourly average mixing ratios of $NO_X$ ($NO_2$) and $SO_2$ at the LAN background station varied in the ranges of 0.4–165.6 ppb (0.2–106.8 ppb) and 0.1–128.6 ppb, respectively. The levels of $NO_X$ and $SO_2$ were highest in winter, followed by spring and autumn, and lowest in summer. Although a significant correlation was observed between the daily average mixing ratios of $NO_X$ and $SO_2$ ($R^2 = 0.29$, $P < 0.001$), their average diurnal variation characteristics differed from each other, with morning peaks in $SO_2$ occurring later than in $NO_X$.

The annual average mixing ratio of $NO_X$ ($NO_2$) fluctuated upwards between 2006 and 2011 (+0.31 ppb/yr, $P = 0.16$) (+0.27 ppb/yr, $P = 0.17$) with a mean value of 13.8 ppb and then began to decrease significantly from 2011 to 2016 (–0.78 ppb/yr, $P < 0.01$) (–0.70 ppb/yr, $P < 0.01$), with a mean value of 13.7 ppb (12.5 ppb). The annual average mixing ratio of $NO_X$ was significantly correlated with the industrial ($R^2 = 0.81$, $P < 0.001$, 2006–2016) and total ($R^2 = 0.88$, $P < 0.001$, 2006–2016) $NO_X$ emissions in the YRD, so as between surface and OMI $NO_2$ (Fig.S3b, $R^2 = 0.61$, $P < 0.01$). The annual 95 % percentile of $NO_X$ mixing ratios followed a similar trend to the annual average, whereas the 5th percentile levels fluctuated upwards at +0.17 ppb/yr from 2006 to 2014, reflecting the increasing regional background level of $NO_X$ in the YRD during those years, which

was related to the continued increase in vehicle numbers in the YRD. The annual average mixing ratio of $SO_2$ exhibited a rapid
and significant decreasing trend (–0.99 ppb/yr, $P < 0.001$) and was closely correlated to total $SO_2$ emission ($R^2 = 0.78$, $P < 0.001$) , total $SO_2$ industrial emission ($R^2 = 0.76$, $P < 0.001$) and total thermal power plants $SO_2$ industrial emission ($R^2 = 0.89$, $P < 0.001$) in the YRD. The reduced emissions were resulted from the strong and effective introduction of national control policies. The yearly decrease of $SO_2/NO_X$. ratios suggest a more effective reduction in $SO_2$ than in $NO_X$. Thus, $NO_X$ emission control needs to be further strengthened in the future.

We found gradual changes in diurnal patterns of both gases. After 2010, both $NO_X$ and $SO_2$ showed diurnal patterns with two peaks and two valleys. The morning peak of $NO_X$ occurred at approximately 09:00, earlier than that of $SO_2$ (10:00), and the evening peak occurred at the same time as $SO_2$ (19:00). The morning and evening peaks of both gases protruded gradually. This phenomenon can hardly be attributed to changes in meteorological conditions (such as the MLD). We believe that changes in major sources of $NO_X$ and $SO_2$ should be the cause, with increasing $NO_X$ emission from vehicles resulting in higher $NO_X$
peaks during rush hours and reduced $SO_2$ emissions from power plants and other large point sources making the $SO_2$ peaks relatively protruding.

*Data availability.* The data of stationary measurements are available upon request to the contact author Weili Lin (linwl@muc.edu.cn).
*Author contributions.* QY wrote the paper, WL and XX developed the idea f, formulated the research goals, and edit the paper. QM and JY carried out the measurement of $NO_X$ and $SO_2$ and analysed the meteorological data.
*Competing interests.* The authors declare that they have no conflict of interest.

**Acknowledgements.**

This study was funded by the National Natural Science Foundation of China (Grant No. 91744206,  21876214).

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

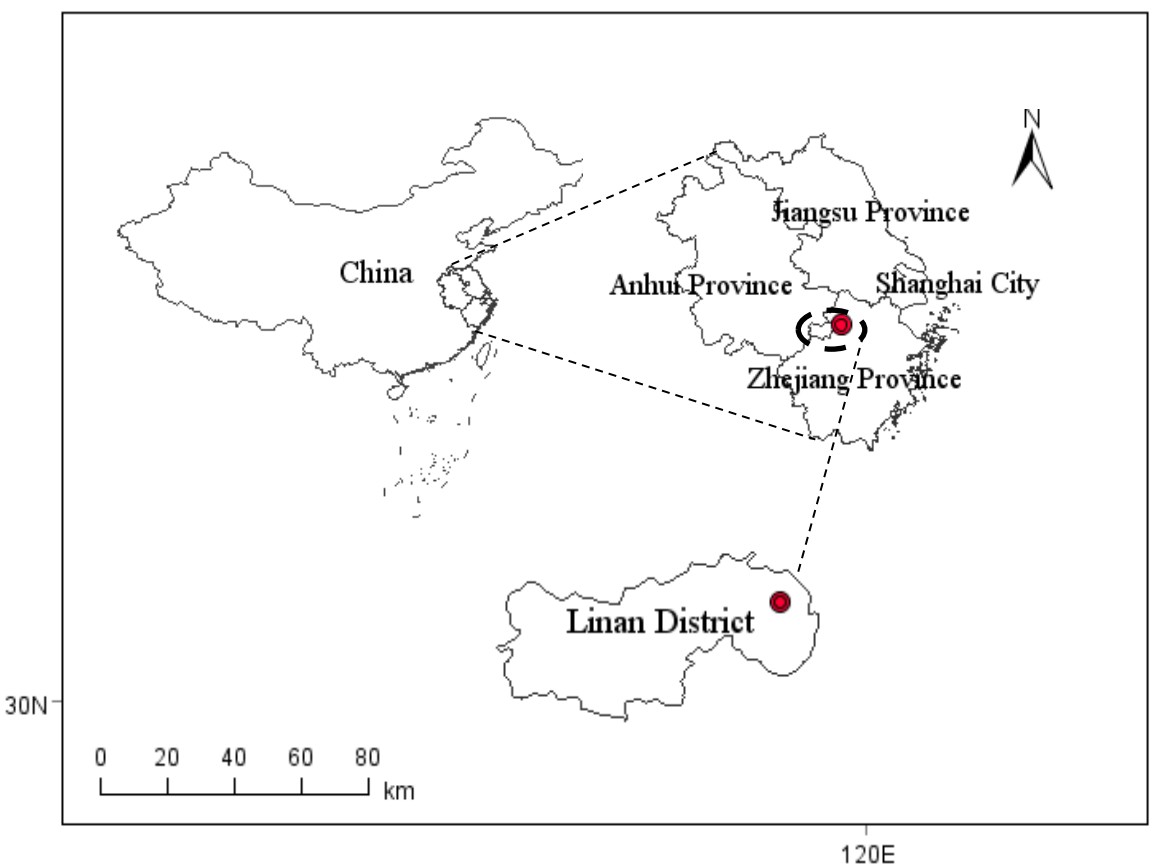


**Figure 1: Geographical location of LAN.**

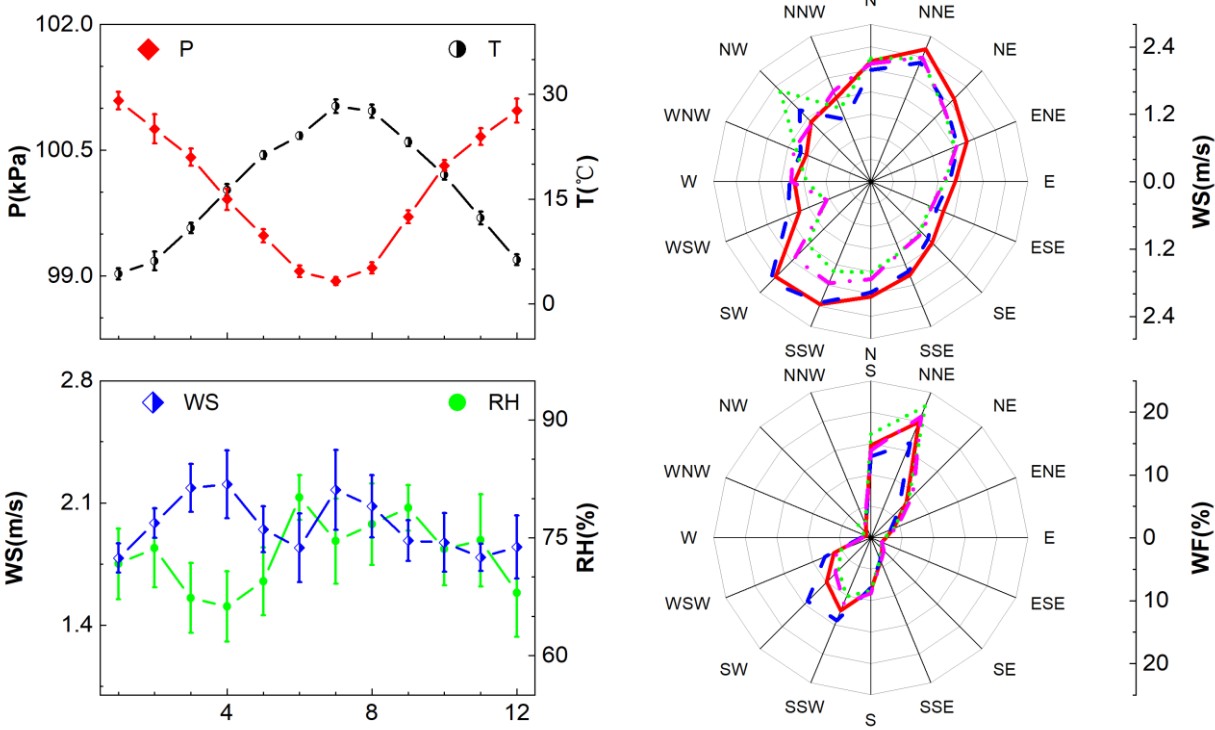

**Figure 2: Average seasonal variations in air pressure (P), temperature (T), wind speed (WS), relative humidity (RH), and rose maps of wind speed (WS) and wind direction frequency (WF) at LAN during 2006–2016. In the rose maps of WS and WF, red solid represents spring, blue dash for summer, green short dot for autumn and magenta dash dot for winter.**

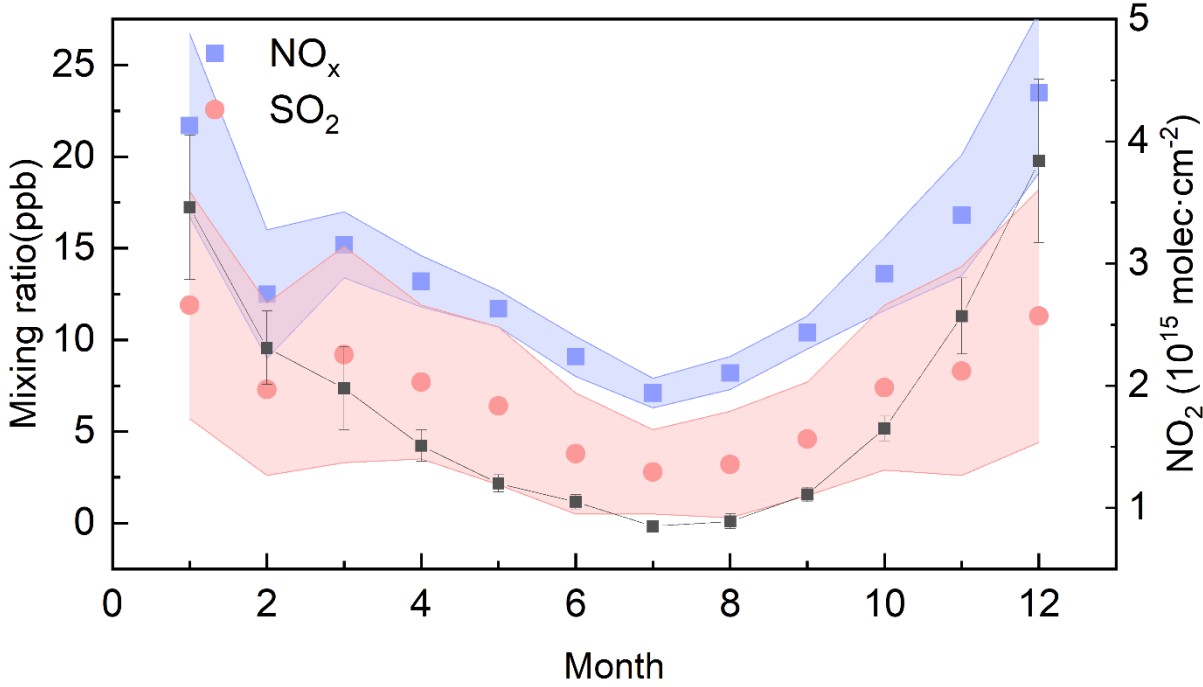

**Figure 3: Monthly average NO$_X$ and SO$_2$ mixing ratios at LAN (left axis) and monthly tropospheric vertical column density of NO$_2$ (right axis) over 115.125° E–122.875° E and 27.125° N–35.875° N in the YRD during 2006–2016.**


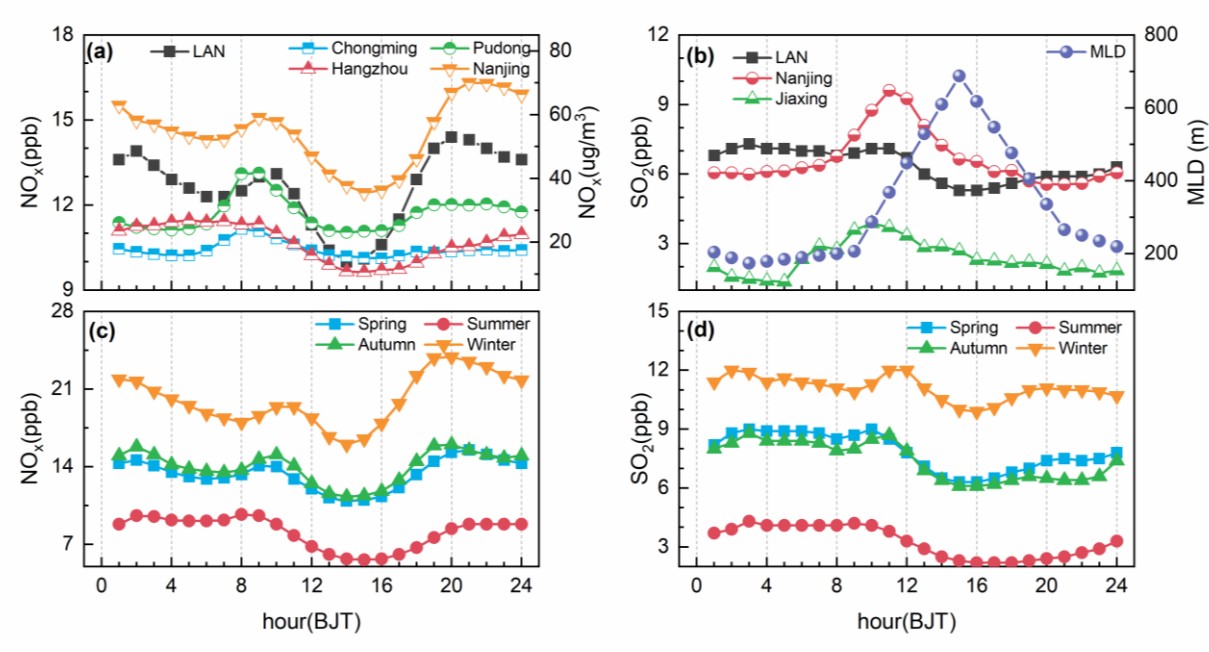

**Figure 4: Annual average diurnal variations in NO$_X$ (a, left axis) and in SO$_2$ (b, left axis) at LAN and its surrounding cities (NO$_X$, a, right axis; SO$_2$, b, left axis); seasonal average diurnal variations in NO$_X$ (c, left axis) and SO$_2$ (d, left axis) at LAN. The average diurnal mixed layer depth (MLD; right axis) is also plotted in panel b.**

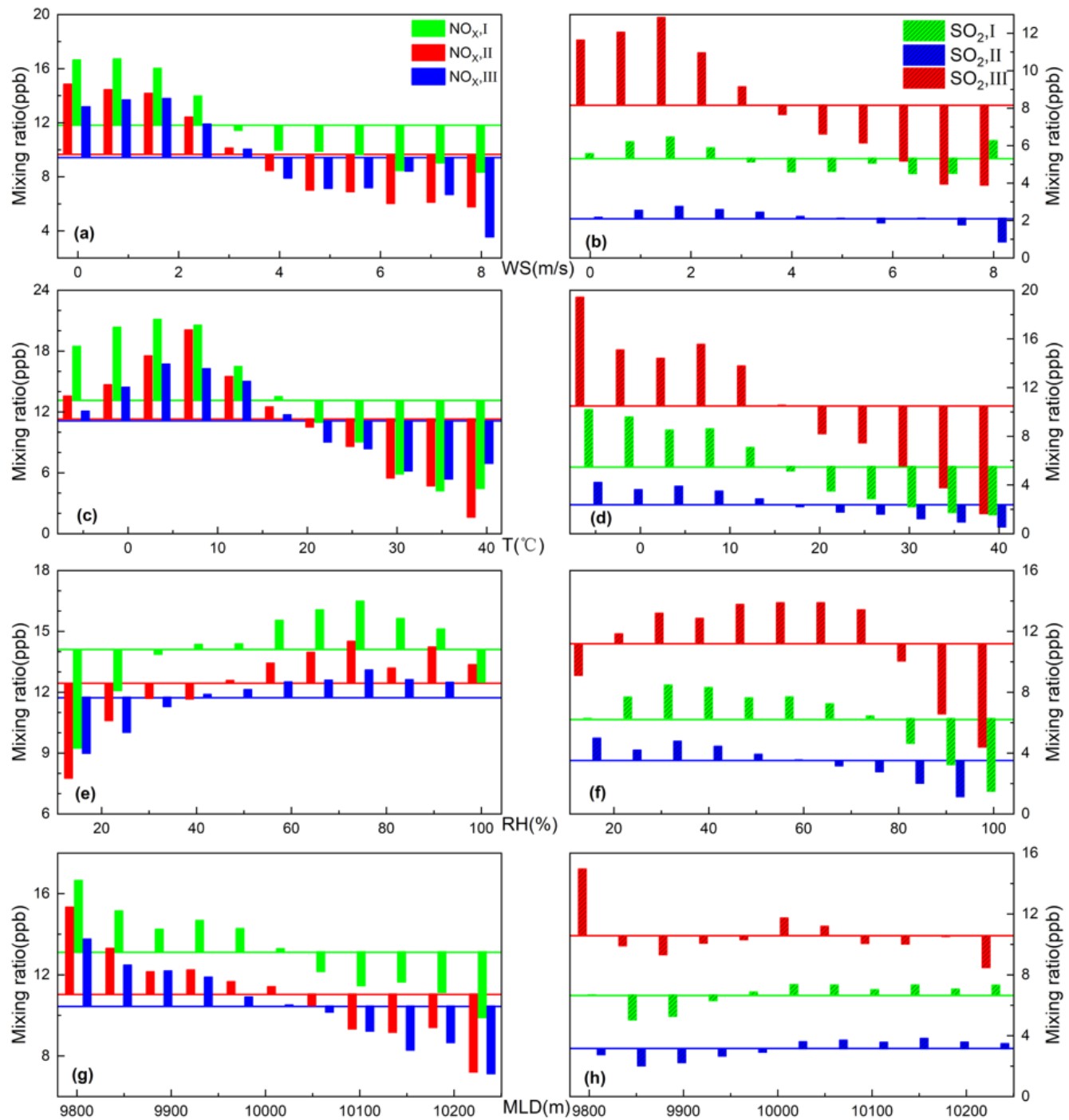

**Figure 5: Variation characteristics of NOX and SO2 with wind speed (WS; a and b), temperature (T; c and d), relative humidity (RH; e and f), and the mixed layer depth (MLD; g and h) at LAN during period I (2006–2009), period II (2010–2013) and period III (2014– 2016). The horizontal lines in the graph indicate the average values of NOX and SO2 for each period. Columns indicate changes relative to the corresponding mean values.**

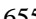

**Figure 6: Seasonal distributions of NOX and SO2 concentrations in different wind directions.**

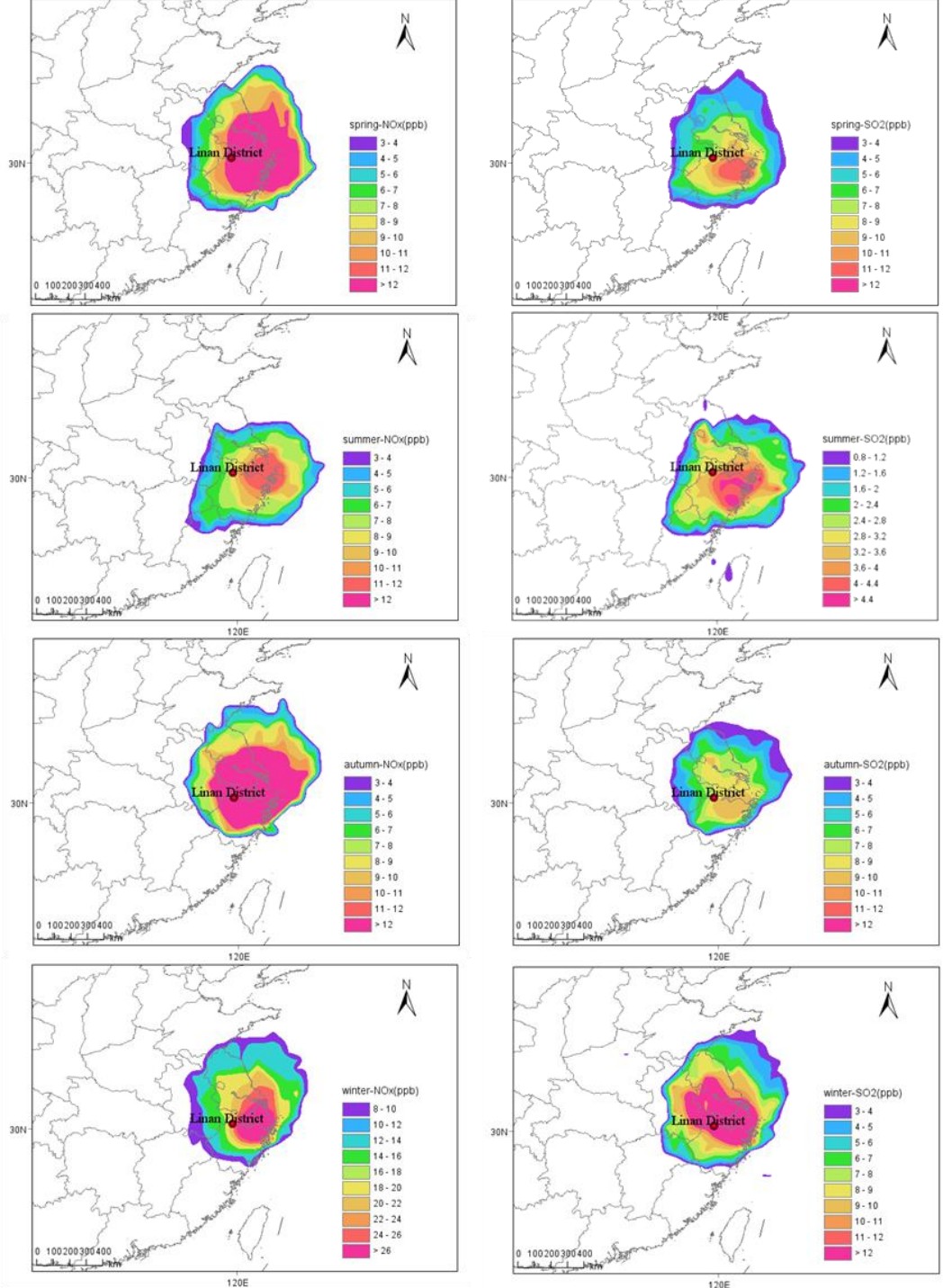

Figure 7: Potential source analysis of NOₓ and SO₂ in different seasons at LAN according to concentration weighted trajectory analysis.

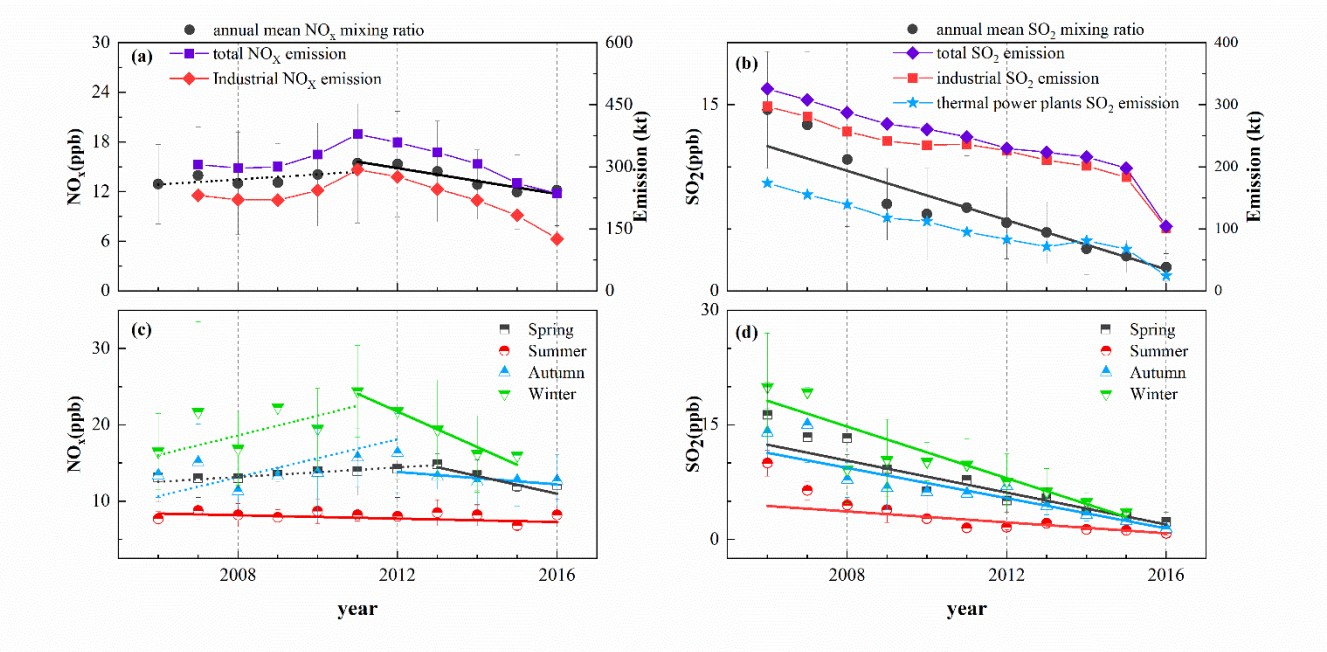

Figure 8: Annual mean $NO_X$ mixing ratio at LAN (left axis) compared with total $NO_X$ emission and industrial $NO_X$ emission in the YRD (a, right axis); annual mean $SO_2$ mixing ratio at LAN (left axis) compared with total $SO_2$ emission, industrial $SO_2$ emission, thermal power plants $SO_2$ emission in the YRD (b, right axis), seasonal average annual variation of $NO_X$ (c), and $SO_2$ (d) at LAN.

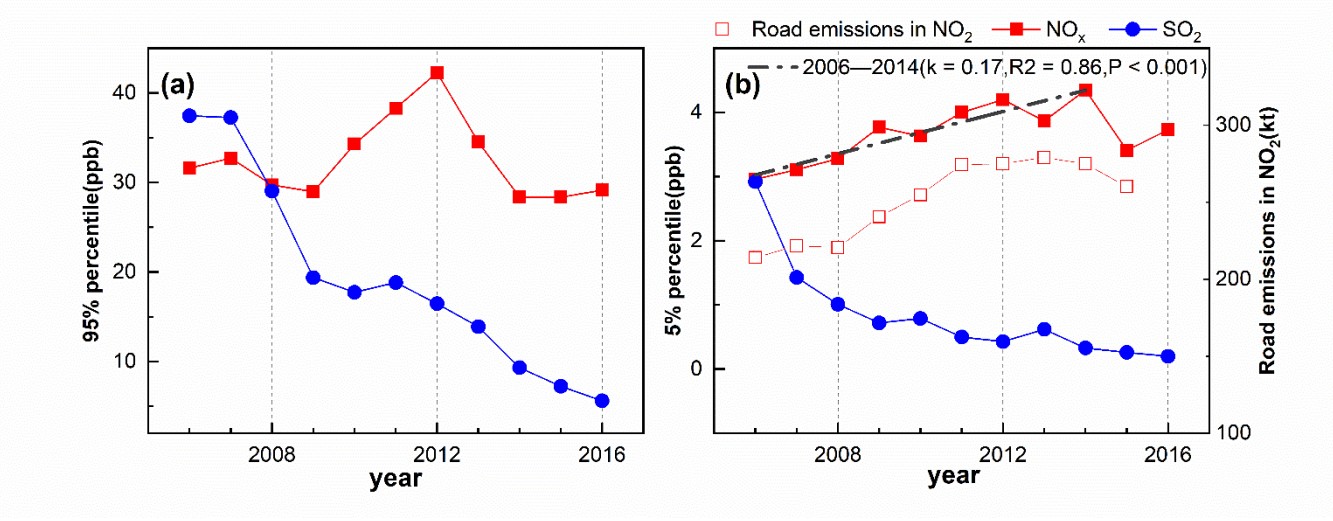

**Figure 9: Annual variations in the 95 % percentile concentration (a) and the 5 % percentile concentration (b) of NOₓ and SO₂ at LAN; data of NO₂ road emissions in the YRD are obtained from the REASv3.2 data sets in the *Regional Emission inventory in Asia* (Kurokawa and Ohara, 2020).**

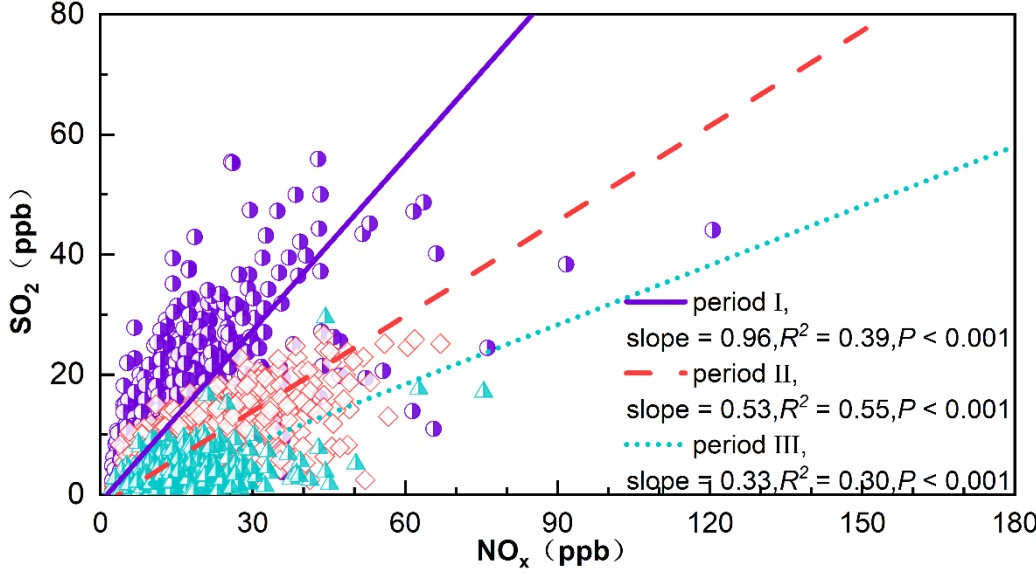

123 **Figure 10: Reduced major axis regressions on the scatter plots of daily average SO₂ and NOₓ mixing ratios during three periods at 675 LAN.**

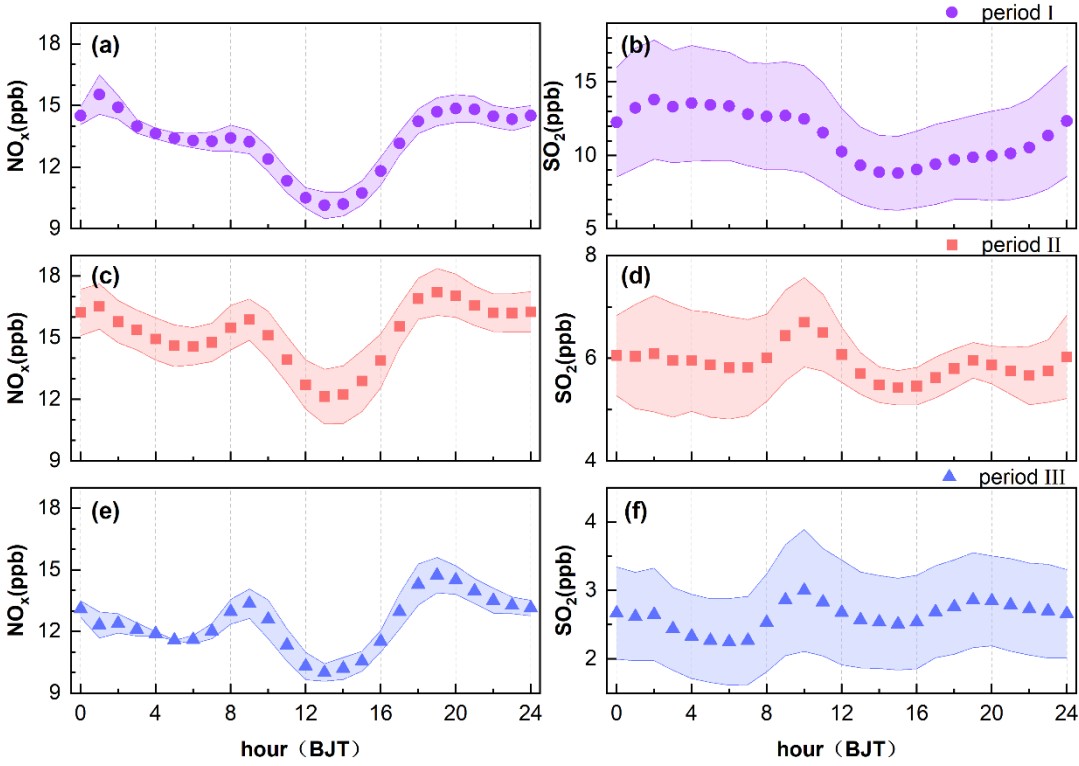

Figure 11: Average diurnal variations in NOX (a, c, e) and in SO2 (b, d, f) during period I (2006–2009), period II (2010–2013) and period III (2014–2016) at LAN.

**Table 1 Statistics of NOx and SO2 levels from 2006 to 2016 at LAN.**

| year | NO2 (ppb) | | | | | NOx (ppb) | | | | | SO2 (ppb) | | | | | SO2/NOx |
|---|---|---|---|---|---|---|---|---|---|---|---|---|---|---|---|---|
| | Ave | Med | SD | Max | Min | Ave | Med | SD | Max | Min | Ave | Med | SD | Max | Min | |
| 2006 | 12.1 | 10.9 | 4.2 | 19.9 | 6.0 | 12.9 | 11.5 | 4.8 | 22.0 | 6.5 | 14.6 | 13.8 | 4.7 | 24.7 | 8.4 | 1.13 |
| 2007 | 12.7 | 11.3 | 5.1 | 24.9 | 6.8 | 13.8 | 11.7 | 6.0 | 29.0 | 7.5 | 13.4 | 12.4 | 5.9 | 23.4 | 5.2 | 0.97 |
| 2008 | 12.0 | 10.8 | 5.0 | 22.5 | 6.2 | 13.0 | 11.3 | 6.2 | 27.9 | 6.6 | 10.6 | 10.6 | 5.4 | 19.9 | 3.7 | 0.82 |
| 2009 | 12.1 | 13.1 | 3.7 | 20.1 | 6.7 | 13.1 | 13.8 | 4.7 | 24.9 | 7.0 | 7.0 | 7.1 | 2.9 | 11.9 | 2.1 | 0.54 |
| 2010 | 12.5 | 11.6 | 5.1 | 24.5 | 6.5 | 14.1 | 12.5 | 6.2 | 29.3 | 7.6 | 6.2 | 5.7 | 3.7 | 14.9 | 1.9 | 0.44 |
| 2011 | 14.1 | 13.0 | 6.0 | 26.5 | 6.7 | 15.4 | 13.8 | 7.2 | 31.3 | 7.5 | 6.7 | 6.7 | 4.2 | 13.7 | 1.1 | 0.44 |
| 2012 | 13.8 | 14.8 | 5.4 | 22.2 | 5.6 | 15.4 | 15.8 | 6.4 | 26.8 | 5.9 | 5.5 | 6.0 | 2.9 | 9.3 | 1.3 | 0.36 |
| 2013 | 13.5 | 12.5 | 5.4 | 23.8 | 6.2 | 14.5 | 13.1 | 6.1 | 27.0 | 6.5 | 4.7 | 4.3 | 2.5 | 10.0 | 1.9 | 0.32 |
| 2014 | 12.1 | 11.8 | 3.7 | 18.8 | 7.0 | 12.9 | 12.4 | 4.2 | 20.2 | 7.3 | 3.4 | 3.0 | 2.1 | 8.6 | 1.0 | 0.26 |
| 2015 | 11.0 | 11.3 | 3.7 | 17.4 | 6.1 | 12.0 | 11.7 | 4.5 | 19.9 | 6.4 | 2.8 | 2.9 | 1.3 | 5.7 | 1.1 | 0.23 |
| 2016 | 11.1 | 10.7 | 3.4 | 16.8 | 6.8 | 12.2 | 11.4 | 4.3 | 19.8 | 7.2 | 1.9 | 1.6 | 1.1 | 3.7 | 0.6 | 0.16 |
| Ave. | 12.5 | 12.0 | 4.6 | 21.6 | 6.4 | 13.6 | 13.1 | 1.2 | 15.4 | 12.0 | 7.0 | 6.2 | 4.2 | 14.6 | 1.9 | 0.52 |

Ave: Average; Med: Median; SD: standard deviation; Max: maximum; Min: minimum.

**Table 2 NOₓ and SO₂ mixing ratios observed at various atmospheric background stations.**

| Station | Latitude and longitude, altitude | Period of observation | NO$_X$/ppb | SO$_2$/ppb | SO$_2$/NO$_X$ | References |
|---|---|---|---|---|---|---|
| Lin'an*, Yangtze River Delta background station | 30.3 °N,119.73 °E, 138 m a.s.l. | 2006.1–2016.12 | 13.6 ± 1.2 | 7.0 ± 4.2 | 0.55 | This study |
| Shangdianzi*, North China Regional Background Station | 40.39°N,117.07°E, 293.9 m a.s.l | 2006.1–2006.12 | 12.7 ± 11.8 | 7.6 ± 10.2 | 0.60 | (Meng et al., 2009) |
| Wuyishan, Eastern China Regional Background Station | 27.58°N,117.72°E, 1139 m a.s.l | 2011.3–2012.2 | 2.70 | 1.48 | 0.55 | (Su et al., 2013) |
| Dinghushan, South China Regional Background Station | 23.2°N,112.5°E, 100m a.s.l | 2009.1–2010.12 | 13.6 | 6.5 | 0.48 | (Chen, 2012) |
| Changbaishan, Northeast China Regional Background Station | 42.4°N,117.5°E, 736 m a.s.l | 2009.1–2010.12 | 4.7 | 2.1 | 0.45 | (Chen, 2012) |
| Fukang, Northwest China Regional Background Station | 44.3°N,87.9°E, 470 m a.s.l | 2009.1–2010.12 | 8.3 | 2.2 | 0.27 | (Chen, 2012) |
| Gonggar Mountain, Southwest China Regional Background Station | 29.92°N,102.61°E, 3541 m a.s.l | 2017.1–2017.12 | 0.90 | 0.19 | 0.21 | (Cheng et al., 2019) |
| Jinsha, Central China Regional Background Station | 29.63°N,114.2°E, 750 m a.s.l | 2006.6–2007.7 | 5.6 ± 5.5 | 2.8 ± 5.5 | 0.5 | (Lin et al., 2011) |

* indicates that the site is also one of the World Meteorological Organization (WMO) Global Atmosphere Watch (GAW/WMO) atmospheric background stations

The date of NO$_2$ all above is converted to NO by a molybdenum NO$_2$-to-NO converter heated to about 325°C.

**Table 3 Pearson correlations among NO$_X$, SO$_2$, and meteorological elements (daily average values).**

|  |  | NO$_x$ | SO$_2$ | WS | T | RH | P | MLD |
|---|---|---|---|---|---|---|---|---|
| NO$_x$ | annual | 1 | 0.54* | −0.25* | −0.47* | −0.01 | 0.42* | −0.06* |
|  | Spring |  | 0.38* | −0.23* | −0.22* | 0.09* | 0.18* | −0.32* |
|  | Summer |  | 0.30* | −0.34* | −0.24* | 0.04 | 0.25* | 0.18* |
|  | Autumn |  | 0.46* | −0.28* | −0.36* | −0.06* | 0.35* | −0.12* |
|  | Winter |  | 0.50* | −0.30* | 0.06 | 0.09* | −0.07* | −0.22* |
| SO$_2$ | annual |  | 1 | −0.09* | −0.34* | −0.41* | 0.39* | 0.08* |
|  | Spring |  |  | −0.05 | −0.04 | −0.41* | 0.17* | −0.05 |
|  | Summer |  |  | 0.00 | 0.07* | −0.32* | 0.11* | −0.02 |
|  | Autumn |  |  | −0.11* | −0.23* | −0.56* | 0.31* | 0.12* |
|  | Winter |  |  | −0.13* | −0.07 | −0.34* | 0.17* | 0.02 |

Two-tailed significance test was used.

*: Significant at 0.05 level of correlation

**Table 4 Annual percentage changes in $NO_X$ and $SO_2$ in various regions.**

| Location | Period | Base year | $NO_X$ | $SO_2$ |
|---|---|---|---|---|
| LAN, this study | 2006–2016 | 2006 | −0.49 %/yr | −8.27 %/yr |
| YRD, China | 2006–2016 | 2006 | −0.45 %/yr | −6.65 %/yr |
| Pearl River Delta, China | 2000–2019 | 2006 | −2.84 %/yr | −3.93 %/yr |
| Wuhan City, China | 2005–2017 | 2006 | +2.08%/yr | −9.46 %/yr |
| North China | 2005–2014 | 2005 | −3.34 %/yr | −0.78 %/yr |
| Northwest China | 2010–2016 | 2010 | +12.98%/yr | −13.06 %/yr |
| New York city in America | 2005–2016 | 2005 | −3.46 %/yr | −5.97 %/yr |
| Kraków city in Poland | 2005–2020 | 2007 | −2.21 %/yr | −3.43 %/yr* |
| Preila station in Lithuania | 2005–2017 | 2006 | −1.60 %/yr | −6.83 %/yr |
| Louis Trichardt in South Africa | 2005–2017 | 2006 | +1.85%/yr | −5.11 %/yr |
| Amersfoort city in South Africa | 2005–2017 | 2006 | +6.50%/yr | +2.95%/yr |