# Peer review of "Measurement report: Long-term variations in surface NOx and SO2 mixing ratios from 2006 to 2016 at a background site in the Yangtze River Delta region, China"

_Atmospheric Chemistry and Physics, 2021_

## Author Response (AR1)

We thank for the constructive comments and suggestions. We revised our manuscript according to the comments and suggestions. The following list the point-to-point response to the comments. The changed texts were highlighted with yellow color.

**Response to comments by referee # 1**

General comments:

This paper reports on $NO_x$ and $SO_2$ measurements at the background site in the Yangtse River Delta region in China.

The site, the instrumental setup, quality control and the data processing procedures have been described in detail. Data are compared to other data from other measurement sites. The long-term trend of both $SO_2$ and $NO_x$, their diurnal and seasonal behavior are discussed and compared to emission data.

There are only few $NO_x$ and $SO_2$ datasets on background sites published and analyzed in depth so far so I would recommend this paper being accepted for publication after the following questions are answered.

Specific comments:

1. Line 85 In the information and methods part, the instrumental setup is described. Here, an essential part is missing. The method to convert NO2 into NO for detection should be given, as well as the method for determining the conversion efficiency. Has gas phase titration been used? Also, it should be mentioned if data were corrected for humidity and ozone effects.

Response: In Model 42C-TL trace-level chemiluminescent analyzer, $NO_2$ is converted to NO by a molybdenum $NO_2$-to-NO converter heated to about 325℃. The converter efficiency was checked annually using gas phase titration (GPT). If the converter efficiency is less than 96%, replace the converter. We add the information in the revised paper. Please see page 3, line 89 for the revision.

2. Line 135 One major concern is that the paper describes trends and seasonal behavior of $NO_x$. However, of the nitrogen oxides, NO2 has the major impact on health. NO2 data should be included into table 1 and discussed. What is the long-term trend of NO2?

Response: In this regional background station, $NO_2$ was the dominant form of $NO_x$, accounting for 82.2 % of $NO_x$, so we didn't present the trends and seasonal behavior of $NO_2$. In the revised manuscript, we included the $NO_2$ in Table 1 and text, but trends and seasonal behavior of $NO_2$ in supplementary material.

3.  Line 167 As the authors point out satellite observations are a valuable tool when analyzing station data. How does the long-term trend of the OMI NO2 observation compare to the $NO_x$ data at the site? Likewise, comparison of in situ observation with station data could help to differentiate between boundary layer effects and emission effects when discussing the diurnal behavior of $NO_x$. How does the diurnal behavior of the satellite observation compare to the diurnal behavior of station data?

Response: Thanks for the suggestion. We think it would be a good idea to compare in situ observation with satellite observations to distinguish between boundary layer effects and emission effects. But, given our current level of knowledge, we need to learn further to complete the work. Here, we compare the monthly average satellite products for $NO_2$ with the monthly mean surface $NO_2$, since OMI covers a point on the ground about once every two days.

4.  Line 207 The pollution roses in Figure 5 show that $SO_2$ and $NO_x$ mixing ratios depend not only on the windspeed but also on the wind direction. Is it possible to add a plot to figure 5 which shows the dependency on wind direction?

Response: Thanks for the suggestion. Rose maps are often used to discuss the relationship between wind direction and pollutant concentration, which details had been plotted in Figure 6 and analyzed in section 3.4.

5.  Line 220 Changes in relative humidity can often by explained with changes in airmasses which are advected from different sites. How does relative humidity change with wind direction? Can the wind direction explain the change of $NO_x$ and $SO_2$ with changing relative humidity?

Response: Thanks for your suggestion. We plot the relative humidity change with wind direction together with $NO_x$ and $SO_2$ rose maps (Figure R1). Also comparing them with figure 1, we think it cannot explain the change of $NO_x$ and $SO_2$ with changing relative humidity.

[Figure]

Figure R1. $NO_x$, $SO_2$, and RH rose maps

6.  Line 237 The authors write that the main source of $SO_2$ and $NO_x$ are east from the site as the show it in figure 7. However, in figure 6 it can be observed that highest mixing ratios were measured with wind coming from west. How can this discrepancy be explained?

Response: Thanks. The wind concentration rose diagram (Figure 6) is a method that can effectively identify short-range transport sources near the ground, while the CWT method (Figure 7) can effectively identify long-range transport sources. Based on a comparison of the two, it can be seen that from the perspective of local emissions, atmospheric $NO_x$ and $SO_2$ at the LAN station are mainly from the northeast and southwest of the station, while long-range transport is influenced by the east.

7.  Line 305 The diurnal behavior is already discussed in chapter 3.3. I would suggest merging chapter 3.3 with lines 305 to 341.

Response: Thanks for the suggestion. Chapter 3.3 aims to provide an overview of the characteristics of the diurnal behavior of $NO_x$ and $SO_2$ over the observation period, while lines 305 to 341 focus on their **long-term** characteristics and causes. So, we kept to separate chapter 3.3 from lines 305 to 341.

8.  Line 327 If the disappearance of the $NO_x$ peak at 1:00 A.M. were due to reduction of industrial emissions, why should industrial emissions peak at 1:00 A.M.? Shouldn´t the effect be seen all over the night?

Response: Yes, $NO_x$ peak at 1:00 AM also puzzled us. A small peak in $NO_x$ and $SO_2$ occurred between 01:00 and 02:00, which might be related to nighttime emissions from unscrupulous enterprises (Fan et al., 2013) or the lower electricity prices after midnight in response to the financial pressure of the 2008 economic crisis and the corresponding increase in electricity prices for industrial users (Sun, 2008). But it's really hard to tell exactly why these small peaks dominate after midnight. These two causes also feel too speculative, so we just present the result here.

We revised the sentences as (Page 11, line 334):

The disappearance of the small peak around 01:00 at night during 2012–2016 may be related to the introduction of stricter air pollution control policies for factories that emit at night. Small peaks in $NO_x$ and $SO_2$ occurred between 01:00 and 02:00, which might be related to nighttime emissions from unscrupulous enterprises (Fan et al., 2013) or more production activities the with lower electricity prices after midnight in response to the financial pressure of the 2008 economic crisis

and the corresponding increase in electricity prices for industrial users (Sun, 2008). In spite of these two reasons, however, it's really hard to tell exactly why these small peaks dominate after midnight.

Fan, Y., Fan, S., Zhang, H., Zu, F., Meng, Q., and He, J.: Characteristics of $SO_2$, $NO_2$, $O_3$ volume fractions and their relationship with weather conditions at Linan in summer and winter, J. Atmos. Sci., 121–128, 2013.

Sun, W.: "A two-pronged approach of "limiting coal prices" and "raising electricity prices, Yangtze River Delta, 58–60, 2008.

9.   Line 337 To my knowledge, the impact of traffic on $SO_2$ emission in China is of minor importance. Have you considered residential sources, which are after industrial emissions and power plant emissions the third most important source of $SO_2$ according to the Multi-resolution Emission Inventory for China (MEIC)?

Response: Thanks. You are right. For the evening peaks, the residential sources should be important for $SO_2$ because it's also in cooking hours. We revised sentence as: "The formation of the evening peaks of $NO_x$ and $SO_2$ may be mainly related to the increase in motor vehicle and residential sources emissions, which are stronger in the rush and cooking hours and that of $SO_2$ may be probably more due to the reduction of power plants emissions." Please see Page 12, line 348 for the revision.

10.   Line 369 Data availability: A link should be provided to where the data are stored in the GAW archive.

Response: As far as I know, the data of $NO_x$ and $SO_2$ data for this site are not available in the GAW archive. The specific reasons are complex.

Technical corrections:
1.   Title: Measurement report: Long-term variations in surface $NO_x$ and $SO_2$ from 2006 to 2016 at a background site in the Yangtze River Delta region, China
Better:
Title: Measurement report: Long-term variations in surface $NO_x$ and $SO_2$ mixing ratios from 2006 to 2016 at a background site in the Yangtze River Delta region, China

Response: Accepted. Please see page 1, line 1 for the revision.

2.   Line 185: The seasonal average diurnal variation in $NO_x$ showed a morning peak of $NO_x$ in summer at 08:00, which is 1 to 2 h earlier than during other seasons (Fig. 4c). This sentence is not clear to me. What is it compared to?

Response: We revised the sentence. "In summer, the seasonal average diurnal variation in $NO_x$ showed a morning peak at 08:00, which time is 1 to 2 h earlier than that occurred in other seasons (Fig. 4c)." Please see Page 7, line 192 for the revision.

3.  Line 221: different periods are well consistent

Response: Accepted. Please see Page 7, line 224 for the revision.

4.  Line 222: A blank is missing

Response: Accepted. Please see Page 8, line 225 for the revision.

5.  Line 250: Please give a reference for the Ecological and Environmental Status Bulletin.

Response: Accepted. Please see Page 8, line 225 for the revision.

6.  Line 254: smaller than those

Response: Accepted. Please see Page 9, line 258 for the revision.

7.  Figure 5: The dependencies of $SO_2$ on meteorological parameters in the figure is blurred from the underlying trend. In figure 5h it cannot be seen if data for 2014-2016 change at all. Maybe it is better to plot changes relative to a mean value.

Response: Accepted. Please see Page 24, figure 5 for the revision.

Figure 6: I would suggest using the same color for $NO_x$ in all the seasons in this plot and label the plots instead.

Response: Accepted. Please see Page 25, figure 6 for the revision.

8.  Figure 11: The different y scales in Figures 11a to Figures 11c makes comparison between the periods difficult. I would suggest using the same scale.

Response: Accepted. Please see Page 29, figure 11 for the revision.

9.  Figure 11: I would suggest naming the periods in the figure caption.

Response: Accepted. Please see Page 29, figure 11 for the revision.

**Response to comments by referee # 2**

General comments

The authors present a concise study on long-term trends of $NO_x$ and $SO_2$ in the region of the Yangtze River Delta based on data acquired at the global GAW station Lin'an.

They interpret diurnal, seasonal and long-term changes and put them in context with trends measured in several other regions or cities in China and across the globe.

They assess correlations with meteorological parameters and emissions by different sources.

To identify source regions a potential source analysis of $NO_x$ and $SO_2$ is applied on the YRD region.

Changes in average diurnal patterns of $NO_x$ and $SO_2$ from 2006 to 2016 where found and attributed to long-term changes of vehicle emissions and industrial emissions, respectively.

Specific comments

l.30, l.307, table 1: Why not denoting it $SO_2/NO_x$ or Sulph./Nitrog.? S/N might be mixed up with signal-to-noise ratio.

Response: Accepted. We use $SO_2/NO_x$ instead of S/N in the revised version.

l.85: For $NO_x$: The method of NO2 to NO conversion is missing and must be explained. Furthermore, were corrections for humidity (respective quenching for CLD technique) and ozone reaction within the inlet line applied on the presented data?

Response: In model 42C-TL trace-level chemiluminescent analyzer, $NO_2$ is converted to NO by a molybdenum $NO_2$-to-NO converter heated to about 325℃. The converter efficiency was checked annually using gas phase titration (GPT). If the converter efficiency is less than 96%, replace the converter. We add the information in the revised paper. Please see Page 3, line 89 for the revision.

l.97: Item (4) might describe more specifically what kind of checks, testings are done and what self-diagnosis does comprise of or even better give a reference where it is explained in more detail.For example, I'm wondering what self-diagnosis means: Is it applying internal thresholds for operating parameters and, if yes, which one and what are the consequences for the instrument or measurements? Is this automatically done by the instrument or by the data acquisition system?

Response: Mostly, the instrument self-diagnosis is applying internal thresholds for operating parameters, which can alert people to carry out manual testing, checking, and maintenance on the instrument. It can't be described in a few words. Technical report from U.S. Environmental Protection Agency can be a reference here. We add a reference of US EPA (2017) in the text.

US EPA: Quality Assurance Handbook for Air Pollution Measurement Systems, Volume II, Ambient Air Quality Monitoring Program, EPA-454/B-17-001, 2017.

l.215: Is it adequate to call it already "effect" if this is an investigation of correlations? Is causality approved yet?

Response: Thanks. We have corrected "effect" to "correlation". Please see Page 8, line 218 for the revision.

l.328: Reference for a change of (specific) air pollution control measures?

Response: You might mean what specific air pollution control measures causing the disappearance of the $NO_x$ peak at 1:00 A.M? PLS see our response to referee #1, question 8.

Technical corrections
Figure 2: P (hPa), factor 10 missing?
Response: Thanks. The factor of 10 is missing. The unit is kPa. Please see Page 21, Figure 2 for the revision.

---

## Editor Decision (ED1)

It should be mentioned that the measurements of $NO_2$ was via conversion to NO by a molybdenum $NO_2$-to-NO converter heated to about 325 °C, which is known to suffer from the interference of other NOy compounds such as PAN and $HNO_3$ (Steinbacher et al., 2007; Jung et al., 2017). This implies that the measured $NO_2$ concentrations have to be viewed as an upper limit. However, it is not possible to quantify the overestimation due to the lack of other information. The interference might be enhanced with the increasing PAN/NOx ratios. Qiu et al. (2020) reported an increasing PAN/$NO_X$ ratio from 2011 to 2018 at a background site in North China Plain, but it is not clear if there was similar increase in PAN/$NO_X$ in the YRD. During the transport of air masses to the background site, $HNO_3$ should have been reduced by deposition and partitioning in the particulate phase and intercepted by filters before NOx is measured. The overestimation of NOx by partial conversion of NOz (NOy-NOx), produced by NOx oxidation, in turn, might be a positive offset in the difference between the concentration and emission of NOx when discussing their long-term trends.

---

## Author Response (AR2)

We thank for the constructive comments and suggestions. We revised our manuscript according to the comments and suggestions. The changes in the revised manuscript are yellow-highlighted. The following are our point-to-point responses to the comments.

1. In the first version of paper the method for conversion of $NO_2$ to NO was not mentioned. In the revised version the authors write that $NO_2$ is converted on heated molybdenum. It is well known that this type of converter produces artefacts which leads to an overestimation of NOx, e.g., (Jung et al., 2017; Steinbacher et al., 2007). Therefore, molybdenum converters should not be used at GAW sites for NO2 conversion (WMO, 2011).

In the paper, the authors should discuss possible interferences caused from this conversion method and their possible consequences with respect to the major findings of their paper.

References:

Jung, J., Lee, J., Kim, B., and Oh, S. (2017). Seasonal variations in the $NO_2$ artifact from chemiluminescence measurements with a molybdenum converter at a suburban site in Korea (downwind of the Asian continental outflow) during 2015–2016, Atmos. Environ., 165, 290-300, doi: https://doi.org/10.1016/j.atmosenv.2017.07.010.

Steinbacher, M., Zellweger, C., Schwarzenbach, B., Bugmann, S., Buchmann, B., Ordóñez, C., Prevot, A. S. H., and Hueglin, C. (2007). Nitrogen oxide measurements at rural sites in Switzerland: Bias of conventional measurement techniques, Journal of Geophysical Research: Atmospheres, 112, doi: https://doi.org/10.1029/2006JD007971.

WMO (2011). WMO/GAW Expert Workshop on Global Long-term Measurements of Nitrogen Oxides and Recommendations for GAW Nitrogen Oxides Network, Geneva, Switzerland, GAW Report 195

Response: Thanks for your kind suggestions. We have also noticed the drawback of this technique, but have to accept what has been available at the site. A favorable $NO_2$ measurement technique based on cavity ring-down principle could be applied in the future. We discuss the possible interference in the revised paper. See page 5, line 146.

"It should be mentioned that the measurements of $NO_2$ was converted to NO by a molybdenum $NO_2$-to-NO converter heated to about 325 ℃, which suffered from the interference of other NOy compounds such as PAN and $HNO_3$ (Steinbacher et al., 2007; Jung et al., 2017). This implies that the measured $NO_2$ concentrations have to be viewed as an upper limit. However, it is not possible to quantify the overestimation due to the lack of other information. The interference might be enhanced with the increasing PAN/NOx ratios. Qiu et al. (2020) reported an increasing PAN/$NO_X$ ratio from 2011 to 2018 at a background site in North China Plain, but it is not clear if there was similar increase in PAN/$NO_X$ in the YRD. During the transport of air masses to the background site, $HNO_3$ should have been reduced by deposition and partitioning in the particulate phase and intercepted by filters before NOx is measured. The overestimation of NOx by partly conversion of NOz (NOy-NOx), which were produced by NOx transformation, in turn, might offset positively the difference between the concentration and emission of NOx when discussing their long-term

trends.*"*

**Reference**

Jung, J., Lee, J., Kim, B., and Oh, S. (2017). Seasonal variations in the NO2 artifact from chemiluminescence measurements with a molybdenum converter at a suburban site in Korea (downwind of the Asian continental outflow) during 2015–2016, Atmos. Environ., 165, 290-300, doi: https://doi.org/10.1016/j.atmosenv.2017.07.010.
Steinbacher, M., Zellweger, C., Schwarzenbach, B., Bugmann, S., Buchmann, B., Ordóñez, C., Prevot, A. S. H., and Hueglin, C. (2007). Nitrogen oxide measurements at rural sites in Switzerland: Bias of conventional measurement techniques, Journal of Geophysical Research: Atmospheres, 112, doi: https://doi.org/10.1029/2006JD007971.
Qiu, Y. L., Ma, Z. Q., Lin, W.L., Quan, W. J., Pu, W.W., Li, Y.R., Zhou, L.Y., Shi, Q.F.: A study of peroxyacetyl nitrate at a rural site in Beijing based on continuous observations from 2015 to 2019 and the WRF-Chem model, Front. Environ. Sci. Eng., 14, 180-190, 2020, https://doi.org/10.1007/s11783-020-1250-0.

2. Line 380:

NOx Data from GAW station should be submitted to the global data archive. For nitrogen oxides data this is the World Data Centre for Reactive Gases (WDCRG) maintained by the Norwegian Institute for Air Research (NILU, https://www.gaw-wdcrg.org/). This is a general requirement for GAW stations; and stations that do not report data to the central database should not be termed GAW stations. On the other hand, I understand that there are different obstacles which take time to be overtaken. So, I can agree with the statement on data availability at this point. Still, I strongly recommend to foster the efforts to submit the data to the World Data Centre

Response: Thank you for your understanding. In fact, the data were shared through other ways, such as the participation in tropospheric ozone assessments. Anyway, we will intensify our efforts to consult with relevant authorities about the submission of data.

Other change:

We added the SO2 and NOx emission data in 2016. See figure 8 and highlight text in the corresponding context.

---

## Author Response (AR3)

We thank for the constructive comments and suggestions. We revised our manuscript according to the comments and suggestions. The changes in the revised manuscript are yellow-highlighted. The following are our responses to the comments.

Comments to the author:

Thank you for the responses to the reviewer's comments, which are sufficient to accept this paper for publication. Please consider the attached technical corrections (English language) in the corrected text on lines 147-157 prior to final publication.

Response: Thanks for your kind suggestions. We have also noticed the drawback of this technique, but have to accept what has been available at the site. A favorable $NO_2$ measurement technique based on cavity ring-down principle could be applied in the future. We discuss the possible interference in the revised paper. See page 5, line 147.

"It should be mentioned that the measurement of $NO_2$ was via conversion to NO by a molybdenum $NO_2$-to-NO converter heated to about 325 °C, which was known to suffer from the interference of other NOy compounds such as PAN and $HNO_3$ (Steinbacher et al., 2007; Jung et al., 2017). This implies that the measured $NO_2$ mixing ratios were higher than actual values. However, it is impossible to quantify the overestimated parts due to the lack of other information. The interference might be enhanced with the increasing ratios of PAN to NOx ($PAN/NO_X$). Qiu et al. (2020) reported an increasing $PAN/NO_X$ from 2011 to 2018 at a background site in the North China Plain, but it is not clear if there was a similar increase in $PAN/NO_X$ in the YRD. During the transport of air masses to the background site, $HNO_3$ should be reduced by deposition or partitioning in the particulate phase and intercepted by filters before NOx was measured. Since NOz (NOy-NOx) was produced by NOx oxidation, the overestimation of NOx by partial conversion of NOz, in turn, might be a positive offset in the difference between the measured mixing ratios and the emission of NOx when discussing their long-term trends."